# LRVS-Fashion: Extending Visual Search with Referring Instructions

**Simon Lepage**[1,2]    **Jérémie Mary**[1]    **David Picard**[2]

[1] CRITEO AI Lab, Paris, France

[2] LIGM, École des Ponts, Marne-la-Vallée, France

{s.lepage, j.mary}@criteo.com   david.picard@enpc.fr

## Abstract

This paper introduces a new challenge for image similarity search in the context of fashion, addressing the inherent ambiguity in this domain stemming from complex images. We present Referred Visual Search (RVS), a task allowing users to define more precisely the desired similarity, following recent interest in the industry. We release a new large public dataset, LRVS-Fashion, consisting of 272k fashion products with 842k images extracted from fashion catalogs, designed explicitly for this task. However, unlike traditional visual search methods in the industry, we demonstrate that superior performance can be achieved by bypassing explicit object detection and adopting weakly-supervised conditional contrastive learning on image tuples. Our method is lightweight and demonstrates robustness, reaching Recall at one superior to strong detection-based baselines against 2M distractors.[1]

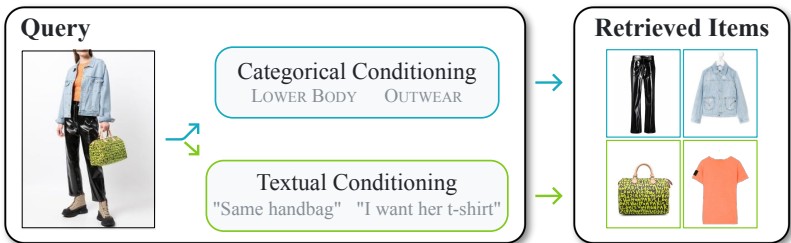

Figure 1: Overview of the Referred Visual Search task. Given a query image and conditioning information, the goal is to retrieve a target instance from a large gallery. *Note that a query is made of an image and an additional text or category, precising what aspect of the image is relevant.*

## 1   Introduction

Image embeddings generated by deep neural networks play a crucial role in a wide range of computer vision tasks. Image retrieval has gained substantial prominence, leading to the development of dedicated vector database systems [22]. These systems facilitate efficient retrieval by comparing embedding values and identifying the most similar images within the database.

Image similarity search in the context of fashion presents a unique challenge due to the inherently ill-founded nature of the problem. The primary issue arises from the fact that two images can be considered similar in various ways, leading to ambiguity in defining a single similarity metric. For

---

[1]The dataset is available at https://huggingface.co/datasets/Slep/LAION-RVS-Fashion

Submitted to the 37th Conference on Neural Information Processing Systems (NeurIPS 2023) Track on Datasets and Benchmarks. Do not distribute.

instance, two images of clothing items may be deemed similar based on their color, pattern, style, or even the model pictured. This multifaceted nature of similarity in fashion images complicates the task of developing a universally applicable similarity search algorithm, as it must account for the various ways in which images can be related.

An intuitive approach is to request users furnish supplementary information delineating their interests, such as providing an image of an individual and denoting interest in the hat (see Fig. 1). Numerous industry leaders including Google, Amazon, and Pinterest have adopted this tactic, however academic discourse on potential alternative methodologies for this task remains scarce as the domain lacks dedicated datasets. For convenience, we propose terming this task Referred Visual Search (RVS), as it is likely to garner attention from the computer vision community due to the utility for product search in extensive catalogs.

In practice, object selection in complex scenes is classically tackled using object detection and crops [21, 17, 12, 42]. Some recent approaches use categorical attributes [8] or text instead [6], and automatically crop the image based on learned attention to input attributes. It is also possible to ask the user to perform the crop himself, yet in all the situations the performance of the retrieval will be sensitive to this extraction step making it costly to build a generic retrieval tool. Recently, Jiao et al. [20] went a step further, incorporating prior knowledge about the taxonomy of fashion attributes and classes without using crops. They use a multi-granularity loss and two sub-networks to learn attribute and class-specific representations, resulting in improved robustness for fashion retrieval, yet without providing any code.

In this work, we seek to support these efforts by providing a dataset dedicated to RVS. We extracted a subset of LAION 5B [41] focused on pairs of images sharing a labeled similarity in the domain of fashion, and propose a method to eliminate the need for explicit detection or segmentation, while still producing similarities in the embedding space specific to the conditioning. We think that such end-to-end approach has the potential to be more generalizable and robust, whereas localization-dependent approaches hinge on multi-stage processing heuristics specific to the dataset.

This paper presents two contributions to the emerging field of Referred Visual Search, aiming at defining image similarity based on conditioning information.

- ✓ The introduction of a new dataset, referred to as LRVS-Fashion, which is derived from the LAION-5B dataset and comprises 272k fashion products with nearly 842k images. This dataset features a test set with an addition of more than 2M distractors, enabling the evaluation of method robustness in relation to gallery size. The dataset's pairs and additional metadata are designed to necessitate the extraction of particular features from complex images.
- ✓ An innovative method for learning to extract referred embeddings using weakly-supervised training. Our approach demonstrates superior accuracy against a strong detection-based baseline and existing published work. Furthermore, our method exhibits robustness against a large number of distractors, maintaining high R@1 even when increasing the number of distractors to 2M.

## 2   Related Work

**Retrieval Datasets.**   Standard datasets in metric learning literature consider that the images are object-centric, and focus on single salient objects [49, 25, 45]. In the fashion domain there exist multiple datasets dedicated to product retrieval, with paired images depicting the same product and additional labeled attributes. A recurrent focus of such datasets is cross-domain retrieval, where the goal is to retrieve images of a given product taken in different situations, for exemple consumer-to-shop [31, 50, 32, 12], or studio-to-shop [32, 27]. The domain gap is in itself a challenge, with issues stemming from irregular lighting, occlusions, viewpoints, or distracting backgrounds. However, the query domain (consumer images for exemple) often contains scenes with multiple objects, making queries ambiguous. This issue has been circumvented with the use of object detectors and landmarks detectors [23, 18, 32, 12]. Some are not accessible anymore [23, 32, 50].

With more than 272k distinct training product identities captured in multi-instance scenes, our new dataset proposes an exact matching task similar to the private Zalando dataset [27], while being larger than existing fashion retrieval datasets and publicly available. We also create an opportunity for new multi-modal approaches, with captions referring to the product of interest in each complex image, and for robustness to gallery size with 2M added distractors at test time.

**Instance Retrieval.** In the last decade, content-based image retrieval has changed because of the arrival of deep learning, which replaced many handcrafted heuristics (keypoint extraction, descriptors, geometric matching, re-ranking...) [11]. In the industry this technology has been of interest to retail companies and search engines to develop visual search solutions, with new challenges stemming from the large scale of such databases. Initially using generic pretrained backbones to extract embeddings with minimal retraining [53], methods have evolved toward domain-specific embeddings supervised by semantic labels, and then multi-task domain-specific embeddings, leveraging additional product informations [58, 3, 46]. The latest developments in the field incorporate multi-modal features for text-image matching [59, 54, 62], with specific vision-language pretext tasks.

However, these methods often consider that the query image is unambiguous, and often rely on a region proposal system to crop the initial image [21, 60, 17, 42, 3, 10]. In our work, we bypass this step and propose an end-to-end framework, leveraging the Transformer architecture to implicitly perform this detection step conditionally to the referring information.

**Referring Tasks.** Referring tasks are popular in vision-language processing, in particular Referring Expression Comprehension and Segmentation where a sentence designates an object in a scene, that the network has to localize. For the comprehension task (similar to open-vocabulary object detection) the goal is to output a bounding box [34, 56, 57, 30]. The segmentation task aims at producing an instance mask for images [61, 34, 19, 7, 24] and recently videos [52, 4]. In this paper, we propose a referring expression task, where the goal is to embed the designated object of interest into a representation that can be used for retrieval. We explore the use of Grounding DINO [30] and Segment Anything [24] to create a strong baseline on our task.

**Conditional Embeddings.** Conditional similarity search has been studied through the retrieval process and the embedding process. On one hand, for the retrieval process, Hamilton et al. [15] propose to use a dynamically pruned random projection tree. On the other hand, previous work in conditional visual similarity learning focused on attribute-specific retrieval, defining different similarity spaces depending on chosen discriminative attributes [47, 36]. They use either a mask applied on the features [47], or different projection heads [36], and require extensive data labeling.

In Fashion, ASEN [35] uses spatial and channel attention to an attribute embedding to extract specific features in a global branch. Dong et al. [8] and Das et al. [6] build upon this model and add a local branch working on an attention-based crop. Recently, Jiao et al. [20] incorporated prior knowledge about fashion taxonomy in this process to create class-conditional embeddings based on known fine-grained attributes, using multiple attribute-conditional attention modules. In a different domain, Asai et al. [1] tackle a conditional document retrieval task, where the user intent is made explicit by concatenating instructions to the query documents. In our work, we use Vision Transformers [9] to implicitly pool features depending on the conditioning information, without relying on explicit ROI cropping or labeled fine-grained attributes.

Composed Image Retrieval (CIR) [48] is another retrieval task where the embedding of an image must be modified following a given instruction. Recent methods use a composer network after embedding the image and the modifying text [28, 5, 2]. While CIR shares similarities with RVS in terms of inputs and outputs, it differs conceptually. Our task focuses on retrieving items based on depicted attributes and specifying a similarity computation method, rather than modifying the image. In Fashion, CIR has been extended to dialog-based interactive retrieval, where an image query is iteratively refined following user instructions [14, 51, 55, 16].

## 3 Dataset

Metric learning methods work by extracting features that pull together images labeled as similar [11]. In our case, we wanted to create a dataset where this embedding has to focus on a specific object in a scene to succeed. We found such images in fashion, thanks to a standard practice in this field consisting in taking pictures of the products alone on neutral backgrounds, and worn by models in scenes involving other clothing items (see Fig. 3).

We created LAION-RVS-Fashion (abbreviated LRVS-F) from LAION-5B by collecting images of products isolated and in context, which we respectively call *simple* and *complex*. We grouped them using extracted product identifiers. We also gathered and created a set of metadata to be used as

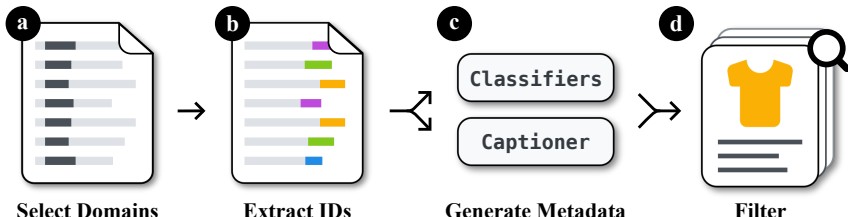

Select Domains     Extract IDs     Generate Metadata     Filter

Figure 2: Overview of the data collection. *a)* Selection of a subset of domains belonging to known fashion retailers. *b)* Extraction of product identifiers in the URLs using domain-specific regular expressions. *c)* Generation of synthetic metadata for the products (categories, captions, ...) using both pretrained and finetuned models. *d)* Deduplication of the images, and assignment to subsets.

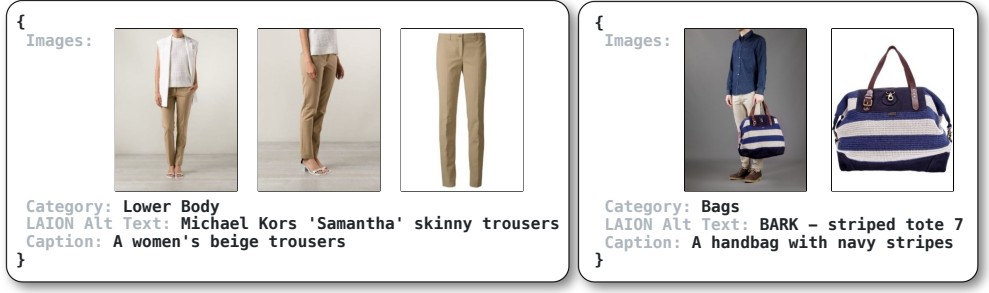

Figure 3: Samples from LRVS-F. Each product is represented on at least a simple and a complex image, and is associated with a category. The simple images are also described by captions from LAION and BLIP2. Please refer to Appendix A.1 for more samples.

referring information, namely LAION captions, generated captions, and generated item categories. The process is depicted Fig. 2, presented in Section 3.1 with additional details in Appendix A.3.

## 3.1 Construction

**Image Collection.** The URLs in LRVS-F are a subset of LAION-5B, curated from content delivery networks of fashion brands and retailers. By analyzing the URL structures we identified product identifiers, which we extracted with regular expressions to recreate groups of images depicting the same product. URLs without distinct identifiers or group membership were retained as distractors.

**Annotations.** We generated synthetic labels for the image complexity, the category of the product, and added new captions to replace the noisy LAION alt-texts. For the complexity labels, we employed active learning to incrementally train a classifier to discern between isolated objects on neutral backdrops and photoshoot scenes. The product categories were formed by aggregating various fine-grained apparel items into 10 coarse groupings. This categorization followed the same active learning protocol. Furthermore, the original LAION captions exhibited excessive noise, including partial translations or raw product identifiers. Therefore, we utilized BLIP-2 [29] to generate new, more descriptive captions.

**Dataset Split.** We grouped together images associated to the same product identifier and dropped the groups that did not have at least a simple and a complex image. We manually selected 400 of them for the validation set, and 2,000 for the test set. The distractors are all the images downloaded previously that were labeled as "simple" but not used in product groups. This mostly includes images for which it was impossible to extract any product identifier.

**Dataset Cleaning.** In order to mitigate false negatives in our results, we utilized Locality Sensitive Hashing and OpenCLIP ViT-B/16 embeddings to eliminate duplicates. Specifically, we removed duplicates between the test targets and test distractors, as well as between the validation targets and validation distractors. Throughout our experiments, we did not observe any false negatives in the results. However, there remains a small quantity of near-duplicates among the distractor images.

## 3.2 Composition

In total, we extracted 272,451 products for training, represented in 841,718 images. This represents 581,526 potential simple/complex positive pairs. We additionally extracted 400 products (800 images) to create a validation set, and 2,000 products (4,000 images) for a test set. We added 99,541 simple images in the validation gallery as distractors, and 2,000,014 in the test gallery.

We randomly sampled images and manually verified the quality of the labels. For the complexity labels, we measured an empirical error rate of $1/1000$ on the training set and $3/1000$ for the distractors. For the product categories, we measured a global empirical error rate of $1\%$, with confusions mostly arising from semantically similar categories and images where object scale was ambiguous in isolated settings (e.g. long shirt vs. short dress, wristband vs. hairband). The BLIP2 captions we provided exhibit good quality, increasing the mean CLIP similarity with the image by $+7.4\%$. However, as synthetic captions, they are not perfect and may contain occasional hallucinations.

Please refer to Appendix A.4 for metadata details, A.5 for considerations regarding privacy and biases and C for metadata details and a datasheet [13].

## 3.3 Benchmark

We define a benchmark on LRVS-F to evaluate different methods on a held-out test set with a large number of distractors. The test set contains 2,000 unseen products, and up to 2M distractors. Each product in the set is represented by a pair of images - a simple one and a complex one. The objective of the retrieval task is to retrieve the simple image of each product from among a vast number of distractors and other simple test images, given the complex image and conditioning information.

For this dataset, we propose to frame the benchmark as an asymmetric task : the representation of simple images (the gallery) should not be computed conditionally. This choice is motivated by three reasons. First, when using precise free-form conditioning (such as LAION texts, which contain hashed product identifiers and product names) a symmetric encoding would enable a retrieval based solely on this information, completely disregarding the image query. Second, for discrete (categorical) conditioning it allows the presence of items of unknown category in the gallery, which is a situation that may occur in distractors. Third, these images only depict a single object, thus making referring information unnecessary. A similar setting is used by Asai et al. [1].

Additionally, we provide a list of subsets sampled with replacement to be used for boostrapped estimation of confidence intervals on the metrics. We created 10 subsets of 1000 test products, and 10 subsets of 10K, 100K and 1M distractors. We also propose a validation set of 400 products with nearly 100K other distractors to monitor the training and for hyperparameter search.

## 4 Conditional Embedding

**Task Formulation.** Let $x_q$ be a query image containing several objects of interest (*e.g.*, a person wearing many different clothes and items), and $c_q$ the associated referring information that provides cues about what aspect of $x_q$ is relevant for the query (*e.g.*, a text describing which garment is of interest, or directly the class of the garment of interest). Similarly, let $x_t$ be a target image, described by the latent information $c_t$. The probability of $x_t$ to be relevant for the query $x_q$ is given by the conditional probability $P(x_t, c_t|x_q, c_q)$. When working with categories for $c_q$ and $c_t$, a filtering strategy consists in assuming independence between the images and their category,

$$P(x_t, c_t|x_q, c_q) = P(x_t|x_q)P(c_t|c_q) , \qquad (1)$$

and further assuming that categories are uncorrelated (*i.e.*, $P(c_t|c_q) = \delta_{c_q=c_t}$ with $\delta$ the Dirac distribution). In this work, we remove those assumptions and instead assume that $P(x_t, c_t|x_q, c_q)$ can be directly inferred by a deep neural network model. More specifically, we propose to learn a flexible embedding function $\phi$ such that

$$\langle \phi(x_q, c_q), \phi(x_t, c_t) \rangle \propto P(x_t, c_t|x_q, c_q) . \qquad (2)$$

Our approach offers a significant advantage by allowing the flexibility to change the conditioning information ($c_q$) at query time, resulting in a different representation that focuses on different aspects of the image. It is also *weakly supervised* in the sense that the referring information $c_q$ is not required

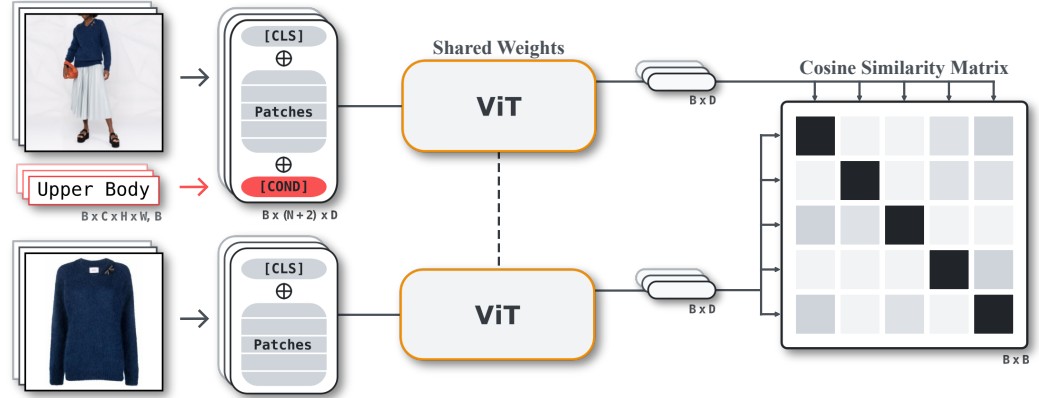

Figure 4: Overview of our method on LRVS-F. For each element in a batch, we embed the scene conditionally and the isolated item unconditionally. We optimize an InfoNCE loss over the cosine similarity matrix. $\oplus$ denotes concatenation to the patch sequence.

to provide localized information about the content of interest (like a bounding box) and can be as imprecise as a free-form text, as shown in Fig. 1.

**Method:** We implement $\phi$ by modifying the Vision Transformer (ViT) architecture [9]. The conditioning is an additional input token with an associated learnable positional encoding, concatenated to the sequence of image patches. The content of this token can either be learned directly (*e.g.* for discrete categorical conditioning), or be generated by another network (*e.g.* for textual conditioning). At the end of the network, we linearly project the [CLS] token to map the features to a metric space. We experimented with concatenating at different layers in the transformer, and found that concatenating before the first layer is the most sensible choice (see Appendix B.1).

We train the network with the InfoNCE loss [44, 38], following CLIP [40], which is detailed in the next paragraph. However, we hypothesize that even though our method relies on a contrastive loss, it does not explicitly require a specific formulation of it. We choose the InfoNCE loss because of its popularity and scalability. During training, given a batch of $N$ pairs of images and conditioning $((x_i^A, c_i^A); (x_i^B, c_i^B))_{i=1..N}$, we compute their conditional embeddings $(z_i^A, z_i^B)_{i=1..N}$ with $z = \phi(x, c) \in \mathbb{R}^d$. We compute a similarity matrix $S$ where $S_{ij} = s(z_i^A, z_j^B)$, with $s$ the cosine similarity. We then optimize the similarity of the correct pair with a cross-entropy loss, effectively considering the $N - 1$ other products in the batch as negatives:

$$l(S) = -\frac{1}{N} \sum_{i=1}^{N} \log \frac{\exp(S_{ii}\tau)}{\sum_{j=1}^{N} \exp(S_{ij}\tau)} \ , \tag{3}$$

with $\tau$ a learned temperature parameter, and the final loss is $\mathcal{L} = l(S)/2 + l(S^\top)/2$. Please refer to Fig. 4 for an overview of the method. The $\tau$ parameter is used to follow the initial formulation of CLIP [40] and is optimized by gradient during the training. At test time, we use FAISS [22] to create a unique index for the entire gallery and perform fast similarity search on GPUs.

## 5 Experiments

We compare our method to various baselines on LRVS-F, using both category- and caption-based settings. We report implementation details before analyzing the results.

### 5.1 Implementation details

All our models take as input images of size $224 \times 224$, and output an embedding vector of 512 dimensions. We use CLIP weights as initialization, and then train our models for 30 epochs with AdamW [33] and a maximum learning rate of $10^{-5}$ determined by a learning rate range test [43]. To avoid distorting pretrained features [26], we start by only training the final projection and new input

embeddings (conditioning and positional) for a single epoch, with a linear warm-up schedule. We then train all parameters for the rest of the epochs with a cosine schedule.

We pad the images to a square with white pixels, before resizing the largest side to 224 pixels. During training, we apply random horizontal flip, and random resized crops covering at least 80% of the image area. We evaluate the Recall at 1 (R@1) of the model on the validation set at each epoch, and report test metrics (recall and categorical accuracy) for the best performing validation checkpoint.

We used mixed precision and sharded loss to run our experiments on multiple GPUs. B/32 models were trained for 6 hours on 2 V100 GPUs, with a total batch size of 360. B/16 were trained for 9 hours on 12 V100, with a batch size of 420. Batch sizes were chosen to maximize GPU memory use.

## 5.2 Results

**Detection-based Baseline**  We leveraged the recent Grounding DINO [30] and Segment Anything [24] to create a baseline approach based on object detection and segmentation. In this setting, we feed the model the query image and conditioning information, which can be either the name of the category or a caption. Subsequently, we use the output crops or masks to train a ViT following the aforementioned procedure. Please refer to Tab. 1 for the results.

Initial experiments conducted with pretrained CLIP features showed a slight preference toward segmenting the object. However, training the image encoder revealed that superior performances can be attained by training the network on crops. Our supposition is that segmentation errors lead to definitive loss of information, whereas the network's capacity is sufficient for it to learn to disregard irrelevant information and recover from a badly cropped image.

Overall, using Grounding DINO makes for a strong baseline. However, it is worth highlighting that the inherent imprecision of category names frequently results in overly large bounding boxes, which in turn limits the performances of the models. Indeed, adding more information into the dataset such as bounding boxes with precise categories would help, yet this would compromise the scalability of the model as such data is costly to obtain. Conversely, the more precise boxes produced by the caption-based model reach $67.8\%$R@1 against 2M distractors.

Table 1: Comparisons of results on LRVS-F for localization-based models. For 0, 10K, 100K and 1M distractors, we report bootstrapped means and standards deviations estimated from 10 randomly sampled sets. We observe superior performances from the caption-based models, due to the precision of the caption which leads to better detections.

| | | Distractors → | +10K | | +100K | | +1M | | +2M | |
|---|---|---|---|---|---|---|---|---|---|---|
| Condi. | Preprocessing | Embedding | %R@1 | %Cat@1 | %R@1 | %Cat@1 | %R@1 | %Cat@1 | %R@1 | %Cat@1 |
| **Category** | Gr. DINO-T + SAM-B | CLIP ViT-B/32 | 16.9 ±1.45 | 67.4 ±1.70 | 8.9 ±0.79 | 65.6 ±1.93 | 4.4 ±0.44 | 64.5 ±1.48 | 2.9 | 64.0 |
| | Gr DINO-T + SAM-B | ViT-B/32 | 83.0 ±1.06 | 94.6 ±0.75 | 69.4 ±1.36 | 92.0 ±0.67 | 53.1 ±1.63 | 90.0 ±0.77 | 46.4 | 89.2 |
| | Gr. DINO-T | ViT-B/32 | 88.7 ±0.74 | 96.4 ±0.55 | 77.0 ±1.79 | 94.3 ±0.82 | 62.8 ±1.92 | 92.2 ±1.26 | 56.0 | 91.8 |
| | Gr. DINO-B | ViT-B/16 | 89.9 ±0.87 | 96.2 ±0.77 | 80.8 ±1.35 | 94.5 ±0.73 | 68.8 ±2.17 | 93.2 ±0.90 | 62.9 | 92.5 |
| **Caption** | Gr. DINO-T + SAM-B | CLIP ViT-B/32 | 27.3 ±1.29 | 72.9 ±1.68 | 16.3 ±0.86 | 71.1 ±1.17 | 9.1 ±0.73 | 70.1 ±1.56 | 6.2 | 69.8 |
| | Gr. DINO-T + SAM-B | ViT-B/32 | 83.5 ±1.56 | 94.6 ±0.39 | 72.2 ±1.59 | 93.0 ±0.42 | 56.5 ±1.61 | 90.9 ±0.74 | 50.8 | 90.2 |
| | Gr. DINO-T | ViT-B/32 | 89.7 ±0.76 | 96.7 ±0.74 | 79.0 ±0.82 | 95.1 ±0.74 | 65.4 ±2.03 | 93.1 ±1.14 | 59.0 | 92.0 |
| | Gr. DINO-B | ViT-B/16 | 91.6 ±0.77 | 97.6 ±0.31 | 83.6 ±0.93 | 96.1 ±0.60 | 73.6 ±1.49 | 94.7 ±0.64 | 67.8 | 94.3 |

**Categorical Conditioning**  We compare our method with categorical detection-based approaches, and unconditional ViTs finetuned on our dataset. To account for the extra conditioning information used in our method, we evaluated the latter on filtered indexes, with only products belonging to the correct category. We did not try to predict the item of interest from the input picture, and instead consider it as a part of the query. We also report unfiltered metrics for reference. Results are in Tab. 2.

Training the ViTs on our dataset greatly improves their performances, both in terms of R@1 and categorical accuracy. Filtering the gallery brings a modest mean gain of $2 - 4\%$R@1 across all quantities of distractors (Fig. 4b), reaching $62.4\%$R@1 for 2M distractors with a ViT-B/16 architecture. In practice, this approach is impractical as it necessitates computing and storing an index for each category to guarantee a consistent quantity of retrieved items. Moreover, a qualitative evaluation of the filtered results reveals undesirable behaviors. When filtering on a category divergent from the network's intrinsic focus, we observe the results displaying colors and textures associated with the automatically focused object rather than the requested one.

Table 2: Comparisons of results on LRVS-F for unconditional, category-based and caption-based models. For 0, 10K, 100K and 1M distractors, we report bootstrapped means and standards deviations from 10 randomly sampled sets. Our CondViT-B/16 outperforms other methods for both groups.

| Distractors → | +10K | | +100K | | +1M | | +2M | |
|---|---|---|---|---|---|---|---|---|
| Model | %R@1 | %Cat@1 | %R@1 | %Cat@1 | %R@1 | %Cat@1 | %R@1 | %Cat@1 |
| ViT-B/32 | 85.6 ±1.08 | 93.7 ±0.31 | 73.4 ±1.35 | 90.9 ±0.78 | 58.5 ±1.37 | 87.8 ±0.86 | 51.7 | 86.9 |
| ViT-B/16 | 88.4 ±0.88 | 94.8 ±0.52 | 79.0 ±1.02 | 92.3 ±0.73 | 66.1 ±1.21 | 90.2 ±0.92 | 59.4 | 88.8 |
| ASEN$_g$ [8] | 63.1 ±1.50 | 76.3 ±1.26 | 46.1 ±1.21 | 68.5 ±0.84 | 29.8 ±1.86 | 62.9 ±1.27 | 24.1 | 62.0 |
| ViT-B/32 + Filt. | 88.9 ±1.01 | — | 76.8 ±1.24 | — | 62.0 ±1.31 | — | 55.1 | — |
| CondViT-B/32 - Category | 90.9 ±0.98 | 99.2 ±0.31 | 80.2 ±1.55 | 98.8 ±0.39 | 65.8 ±1.42 | 98.4 ±0.65 | 59.0 | 98.0 |
| ViT-B/16 + Filt. | 90.9 ±0.88 | — | 81.9 ±0.87 | — | 68.9 ±1.11 | — | 62.4 | — |
| CondViT-B/16 - Category | 93.3 ±1.04 | 99.5 ±0.25 | 85.6 ±1.06 | 99.2 ±0.35 | 74.2 ±1.82 | 99.0 ±0.42 | 68.4 | 98.8 |
| CoSMo [28] | 88.3 ±1.30 | 97.6 ±0.45 | 76.1 ±1.85 | 96.0 ±0.32 | 59.1 ±1.42 | 94.7 ±0.40 | 52.1 | 94.8 |
| CLIP4CIR [2] | 92.9 ±0.64 | 99.0 ±0.33 | 81.9 ±1.63 | 98.1 ±0.68 | 66.9 ±2.05 | 96.5 ±0.67 | 59.1 | 95.5 |
| CondViT-B/32 - Caption | 92.7 ±0.77 | 99.1 ±0.30 | 82.8 ±1.22 | 98.7 ±0.40 | 68.4 ±1.50 | 98.1 ±0.43 | 62.1 | 98.0 |
| CondViT-B/16 - Caption | 94.2 ±0.90 | 99.4 ±0.37 | 86.4 ±1.13 | 98.9 ±0.49 | 74.6 ±1.65 | 98.4 ±0.58 | 69.3 | 98.2 |

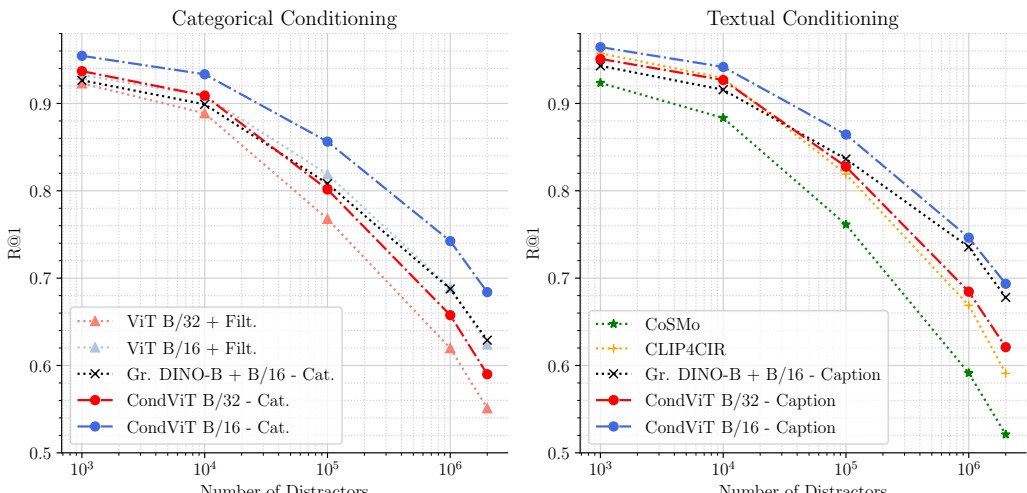

Figure 5: R@1 with repects to number of added distractors, evaluated on the entire test set. Please refer to Tab. 1 and 2 for bootstrapped metrics and confidence intervals. Our categorical CondViT-B/16 reaches the performances of the best caption-based models, while using a sparser conditioning.

We also compare with ASEN [8] trained on our dataset using the authors' released code. This conditional architecture uses a global and a local branch with conditional spatial attention modules, respectively based on ResNet50 and ResNet34 backbones, with explicit ROI cropping. However in our experiments the performances decrease with the addition of the local branch in the second training stage, even after tuning the hyperparameters. We report results for the global branch.

We train our CondViT using the categories provided in our dataset, learning an embedding vector for each of the 10 clothing categories. For the $i$-th product in the batch, we randomly select in the associated data a simple image $x_s$ and its category $c_s$, and a complex image $x_c$. We then compute their embeddings $z_i^A = \phi(x_c, c_s), z_i^B = \phi(x_s)$. We also experimented with symmetric conditioning, using a learned token for the gallery side (see Appendix B.1).

Our categorical CondViT-B/16, with $68.4\%$R@1 against 2M distractors significantly outperforms all other category-based approaches (see Fig. 5, left) and maintains a higher categorical accuracy. Furthermore, it performs similarly to the detection-based method conditioned on richer captions, while requiring easy-to-aquire coarse categories. It does so without making any assumption on the semantic nature of these categories, and adding only a few embedding weights (7.7K parameters) to the network, against 233M parameters for Grounding DINO-B. We confirm in Appendix B.2 that its attention is localized on different objects depending on the conditioning.

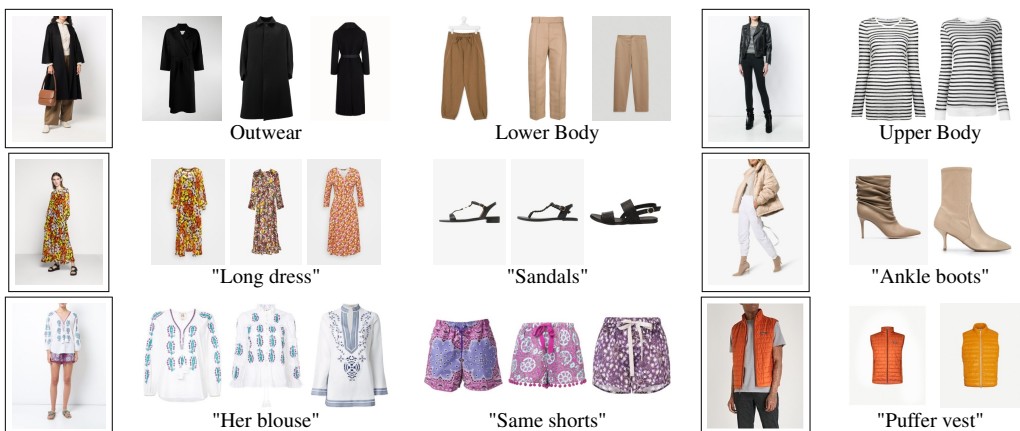

Figure 6: Qualitative results for our categorical (first 2 rows) and textual (last 2 rows) CondViT-B/16. We use free-form textual queries instead of BLIP2 captions to illustrate realistic user behavior, and retrieve from the whole test gallery. See Fig. 13 and 14 in the Appendix for more qualitative results.

**Textual Conditioning**   To further validate our approach, we replaced the categorical conditioning with referring expressions, using our generated BLIP2 captions embedded by a Sentence T5-XL model [37]. We chose this model because it embeds the sentences in a 768-dimensional vector, allowing us to simply replace the categorical token. We pre-computed the caption embeddings, and randomly used one of them instead of the product category at training time. At test time, we used the first caption.

In Tab. 2, we observe a gain of 3.1%R@1 for the CondViT-B/32 architecture, and 0.9%R@1 for CondViT-B/16, compared to categorical conditioning against 2M distractors, most likely due to the additional details in the conditioning sentences. When faced with users, this method allows for more natural querying, with free-form referring expressions. See Figure 6 for qualitative results.

We compare these models with CIR methods: CoSMo [28] and CLIP4CIR [2]. Both use a compositor network to fuse features extracted from the image and accompanying text. CoSMo reaches performances similar to an unconditional ViT-B/32, while CLIP4CIR performs similarly to our textual CondViT-B/32. We hypothesize that for our conditional feature extraction task, early conditioning is more effective than modifying embeddings through a compositor at the network's end. Our CondViT-B/16 model significantly outperforms all other models and achieves results comparable to our caption-based approach using Grounding DINO-B (see Fig. 5, right). As the RVS task differs from CIR, despite both utilizing identical inputs, this was anticipated. Importantly, CondViT-B/16 accomplishes this without the need for explicit detection steps or dataset-specific preprocessing. Notably, we observe that our models achieve a categorical accuracy of 98% against 2M distractors, surpassing the accuracy of the best corresponding detection-based model, which stands at 94.3%.

# 6   Conclusion & Limitations

We studied an approach to image similarity in fashion called Referred Visual Search (RVS), which introduces two significant contributions. Firstly, we introduced the LAION-RVS-Fashion dataset, comprising 272K fashion products and 842K images. Secondly, we proposed a simple weakly-supervised learning method for extracting referred embeddings. Our approach outperforms strong detection-based baselines. These contributions offer valuable resources and techniques for advancing image retrieval systems in the fashion industry and beyond.

However, one limitation of our approach is that modifying the text description to refer to something not present or not easily identifiable in the image does not work effectively. For instance, if the image shows a person carrying a green handbag, a refined search with "red handbag" as a condition would only retrieve a green handbag. The system may also ignore the conditioning if the desired item is small or absent in the database. Examples of such failures are illustrated in Appendix B.3. Additionally, extending the approach to more verticals would be relevant.

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
