# Appendix

# A   Dataset

## A.1   Samples

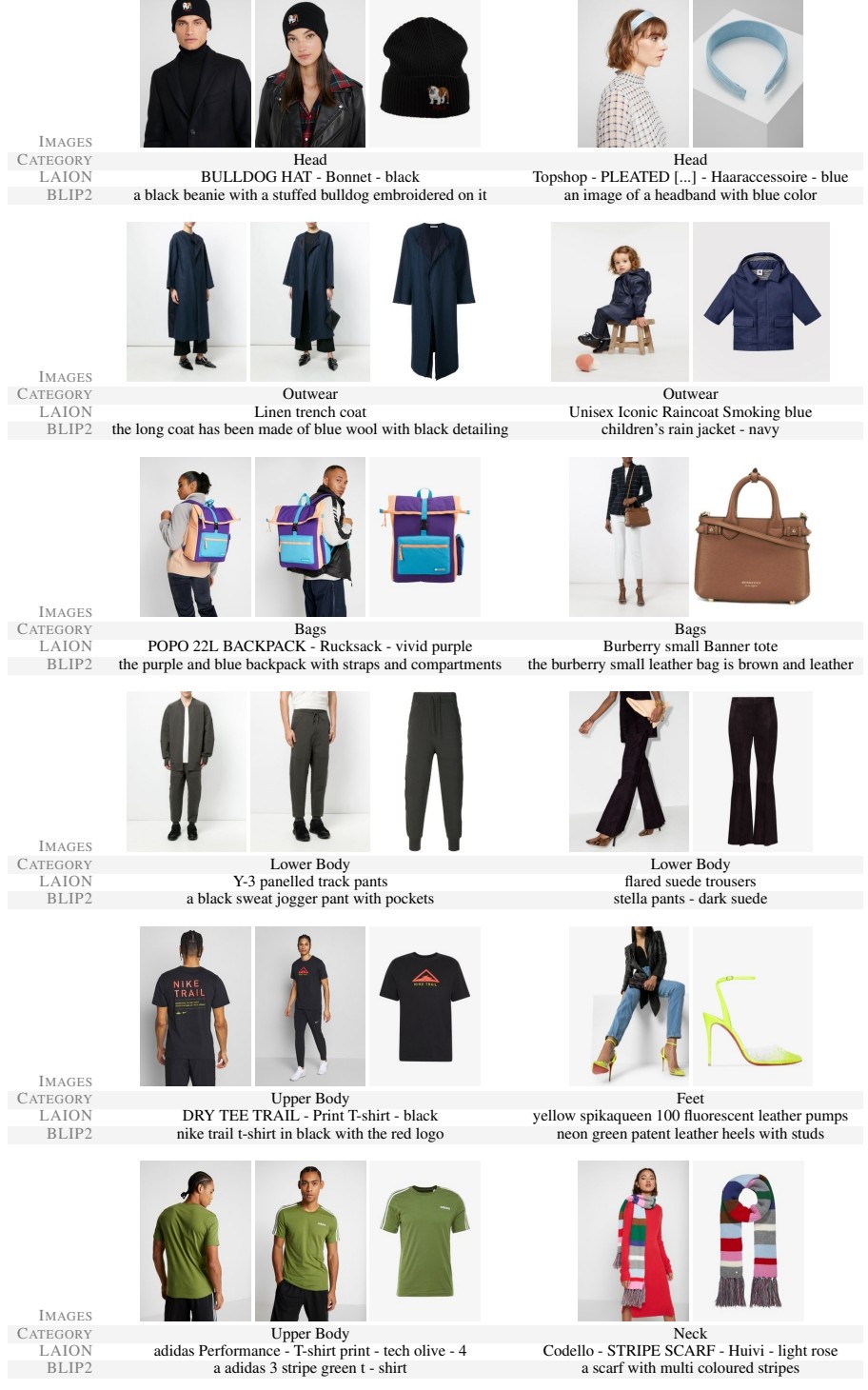

Figure 7: Additional samples from LRVS-F.

## A.2 Usage

To ensure the reproducibility of our work, we release the assets and tools that we created:

- Full Dataset: `https://huggingface.co/datasets/Slep/LAION-RVS-Fashion`
- Test set: `https://zenodo.org/doi/10.5281/zenodo.11189942`
- Training Code: `https://github.com/Simon-Lepage/CondViT-LRVSF`
- Evaluation Code: `https://github.com/Simon-Lepage/LRVSF-Benchmark`
- Categorical Model: `https://huggingface.co/Slep/CondViT-B16-cat`
- Textual Model: `https://huggingface.co/Slep/CondViT-B16-txt`
- Leaderboard: `https://huggingface.co/spaces/Slep/LRVSF-Leaderboard`
- Demo: `https://huggingface.co/spaces/Slep/CondViT-LRVSF-Demo`

The dataset is hosted by Huggingface, using the widely used parquet format, released under the *CC-BY-NC-4.0* license, for research use only. The images can easily be downloaded using tools like `img2dataset`.

## A.3 Construction

**Image Collection:** The raw data of LRVS-F are collected from a list of fashion brands and retailers whose content delivery network domains were found in LAION 5B. We used the automatically translated versions of LAION 2B MULTI and LAION 1B NOLANG to get english captions for all the products. This represents around 8M initial images.

We analyzed the format of the URLs for each domain, and extracted image and product identifiers using regular expressions when possible. We removed duplicates at this step using these identifiers, and put aside images without clear identifiers to be filtered and used as distractors later.

**Image Annotation:** The additional metadata that we provide were generated using deep learning models. We generated indicators of the image complexity, classified the products in 11 categories, and added new image captions.

First, we used a model to classify the complexity of the images, trained with active learning. We started by automatically labeling a pool of images using information found in the URLs, before manually filtering the initial data, and splitting between training and validation. Then, we computed and stored the pre-projection representations extracted by OpenCLIP B16 for each image, and trained a 2-layers MLP to predict the category. After training, we randomly sampled 1000 unlabeled images and annotated the 100 with the highest prediction entropy, before splitting them between training and validation data. We repeated these 2 steps until reaching over 99% accuracy and labeled the entire dataset using this model.

We used a second model to automatically assign categories to the simple images. LAION captions are noisy, so instead of using them we used BLIP2 FlanT5-XL [29] to answer the question "In one word, what is this object?". We gathered all the nouns from the answers, using POS tagging when the generated answer was longer, and grouped them in 11 categories (10 for clothing, 1 for non-clothing). We automatically created an initial pool of labeled data, which we manually filtered, before applying the same active learning process as above. We then annotated all the simple images with this model. Please refer to Appendix A.4 for the list of categories and their composition.

Finally, we automatically added new descriptions to the simple images, because the quality of some LAION texts was low. For example, we found partially translated sentences, or product identifiers. We generated 10 captions for each image using BLIP2 FlanT5-XL with nucleus sampling, and kept the two with largest CLIP similarity.

**Dataset Split:** We grouped together images associated to the same product identifier and dropped the groups that did not have at least a simple and a complex image. We manually selected 400 of them for the validation set, and 2,000 for the test set. The distractors are all the images downloaded previously that were labeled as "simple" but not used in product groups. This mostly includes images for which it was impossible to extract any product identifier.

Finally, we used Locality Sensitive Hashing (LSH) with perceptual hash, and OpenCLIP B16 embeddings to remove duplicates. We created FAISS indexes based respectively on hamming distance and cosine similarity, automatically removing samples with extremely high similarity. We manually inspected samples near the threshold. We used this process on complex images from the training set to remove products duplicates, on train and test sets to reduce evaluation bias, and on gallery images and distractors for both the validation and test sets.

## A.4 Composition

We classified LRVS-F products into 11 distinct categories. Among these categories, 10 are specifically related to clothing items, which are organized based on their approximate location on the body. Additionally, there is one non-clothing category included to describe some distractors. Tab. 3 provides information regarding the counts of products within each category, as well as the data split. For a more detailed understanding of the clothing categories, Tab. 4 presents examples of fine-grained clothing items that are typically associated with each category.

Each product in our dataset is associated with at least one simple image and one complex image. In Figure 8, we depict the distribution of simple and complex images for each product. Remarkably, we observe that the majority of products, accounting for 90% of the dataset, possess a single simple image and up to four complex images.

Table 3: Count of simple images (isolated items) across the dataset splits. Some training products are depicted in multiple simple images, hence the total higher than the number of unique identities.

| | Upper Body | Lower Body | Whole Body | Outwear | Bags | Feet | Neck | Head | Hands | Waist | NonClothing | Total |
|---|---|---|---|---|---|---|---|---|---|---|---|---|
| Train | 92 410 | 75 485 | 48 446 | 45 867 | 26 062 | 4 224 | 3 217 | 1 100 | 190 | 184 | - | 297 185 |
| Val | 80 | 80 | 80 | 80 | 60 | 6 | 6 | 4 | 2 | 2 | - | 400 |
| Test | 400 | 400 | 400 | 400 | 300 | 30 | 30 | 20 | 10 | 10 | - | 2 000 |
| Val. Dist. | 19 582 | 13 488 | 8 645 | 6 833 | 10 274 | 22 321 | 2 470 | 6 003 | 2 866 | 1 016 | 6 043 | 99 541 |
| Test Dist. | 395 806 | 272 718 | 172 385 | 136 062 | 203 390 | 448 703 | 50 881 | 121 094 | 57 271 | 19 853 | 121 851 | 2 000 014 |

Table 4: Examples of sub-categories.

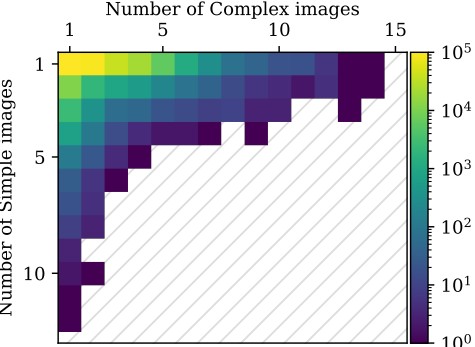

| CATEGORY | COMPOSITION |
|---|---|
| Upper Body | T-shirts, Shirts, Crop Tops, Jumper, Sweater … |
| Lower Body | Shorts, Pants, Leggings, Skirts … |
| Whole Body | Dress, Gown, Suits, Rompers … |
| Outwear | Coat, Jacket … |
| Bags | Handbags, Backpack, Luggage … |
| Feet | Shoes, Boots, Socks … |
| Neck | Scarves, Necklace … |
| Head | Hat, Cap, Glasses, Sunglasses, Earrings … |
| Hands | Gloves, Rings, Wristbands… |
| Waist | Belts |

Figure 8: Distribution of Simple and Complex images across products. 90% of the products have 1 simple image and up to 4 complex images.

## A.5 Ethics Statement

**Harmful and Private Content.** Our dataset is a subset of the publicly released LAION 5B dataset, enriched with synthetic metadatas (categories, captions, product identifiers). However, our process began by curating a subset of domains, focusing exclusively on domains affiliated with well-known fashion retailers and URLs containing product identifiers. As such, these images come from large commercial fashion catalogs. Our dataset contains images that appear in online fashion catalogs and does not contain harmful or disturbing images. Most of the images are pictures of isolated attire on neutral backgrounds. Images depicting people are all extracted from professional photoshoots, with all the ethical and legal considerations that are common practices in the fashion retail industry.

We release our dataset only for research purposes as a benchmark to study Referred Visual Search where no public data exists, which is a problem for reproducibility. This is an object-centric instance retrieval task that aims to control more precisely the content of image embeddings. On this dataset, to optimize the performances, embeddings should only contain information regarding the referred garment, rather than the model wearing it.

**Dataset Biases.** Our dataset lacks metadata for a comprehensive exploration of bias across gender and ethnicity. However, based on an inspection of a random sample of 1000 images, we estimate that roughly 2/3 of the individuals manifest discernible feminine physical attributes or attire.

Among the cohort of 22 fashion retailers featured in our dataset, 14 are from the European Union, 7 are from the United States, and the remaining one is from Russia. Thereby, even though these retailers export and sell clothing across the world, our dataset reproduces the biases of European and American fashion industries with respect to models' ethnicity and gender.

# B Model

## B.1 Ablation Studies

**Insertion Depth.** We study the impact of the insertion depth of our additional conditioning token by training a series of CondViT-B/32, concatenating the conditioning token before different encoder blocks for each one of them.

Fig. 9 indicates that early concatenation of the conditioning token is preferable, as we observed a decrease in recall for deep insertion (specifically, layers 10-12). However, there was no statistically significant difference in performance between layers 1-8. Consequently, we decided to concatenate the token at the very beginning of the model. We hypothesize that the presence of residual connections in our network enables it to disregard the conditioning token until it reaches the optimal layer. The choice of this layer may depend on factors such as the size of the ViT model and the characteristics of the dataset being used.

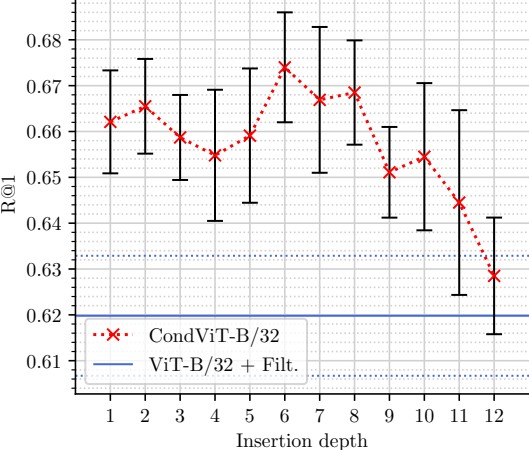

Figure 9: R@1 on the test set with respect to the insertion depth of the conditioning token. Error bars represent the bootstrapped estimation of the standard deviation across 10 splits. Late insertion degrades performance, but no significant difference can be seen among the first layers.

**Asymmetric Conditioning.** We experiment with using conditioning for the simple images too, using a single learned "empty" token for all the simple images. We denote this token $c_\emptyset$. Then for each simple image $x_s$ we compute its embedding as $\phi(x_s, c_\emptyset)$.

Results in Tab. 5 show that there is no really significant difference between both approaches, even though CondViT-B/16 results are better without this additional token for large amounts of distractors ($\geq$ 100K). We choose to keep an asymmetric embedding process.

Table 5: Comparison of symmetric and asymmetric conditioning on LRVS-F test set. We report bootstrapped mean and standard deviation on the test set. There is no significant difference between the configurations. Bold results indicate a difference of more than $1\%$.

| Distractors → | +0 | | +10K | | +100K | | +1M | | +2M | |
|---|---|---|---|---|---|---|---|---|---|---|
| Model | %R@1 | %Cat@1 | %R@1 | %Cat@1 | %R@1 | %Cat@1 | %R@1 | %Cat@1 | %R@1 | %Cat@1 |
| CondViT-B/32 | 97.0 ±0.57 | 100 ±0.07 | 90.9 ±0.98 | 99.2 ±0.31 | 80.2 ±1.55 | 98.8 ±0.39 | 65.8 ±1.42 | 98.4 ±0.65 | 59.0 | 98.0 |
| CondViT-B/32 + $c_\emptyset$ | 96.8 ±0.94 | 100 ±0.10 | 91.1 ±1.04 | 99.3 ±0.24 | 79.9 ±1.35 | 99.0 ±0.21 | 66.0 ±1.36 | 98.3 ±0.46 | 59.6 | 98.2 |
| CondViT-B/16 | 97.7 ±0.21 | 99.8 ±0.12 | 93.3 ±1.04 | 99.5 ±0.25 | **85.6** ±1.06 | 99.2 ±0.35 | **74.2** ±1.82 | 99.0 ±0.42 | **68.4** | 98.8 |
| CondViT-B/16 + $c_\emptyset$ | 97.8 ±0.32 | 99.9 ±0.11 | 93.2 ±0.79 | 99.5 ±0.16 | 84.4 ±1.16 | 99.0 ±0.29 | 72.5 ±1.88 | 98.8 ±0.42 | 66.5 | 98.0 |

## B.2 Attention Maps

We propose a visualization of the attention maps of our ViT-B/16, ASEN, and our categorical CondViT-B/16 in Fig. 10. We compare attention in the last layer of the transformers with the Spatial Attention applied at the end of ASEN's global branch. We observe that the attention mechanism in the transformers exhibits a notably sparse nature, selectively emphasizing specific objects within the input scene. Conversely, ASEN demonstrates a comparatively less focused attention distribution. Surprisingly, the unconditional ViT model exhibits a strong focus on a single object of the scene, while the attention of our CondViT dynamically adjusts in response to the conditioning information.

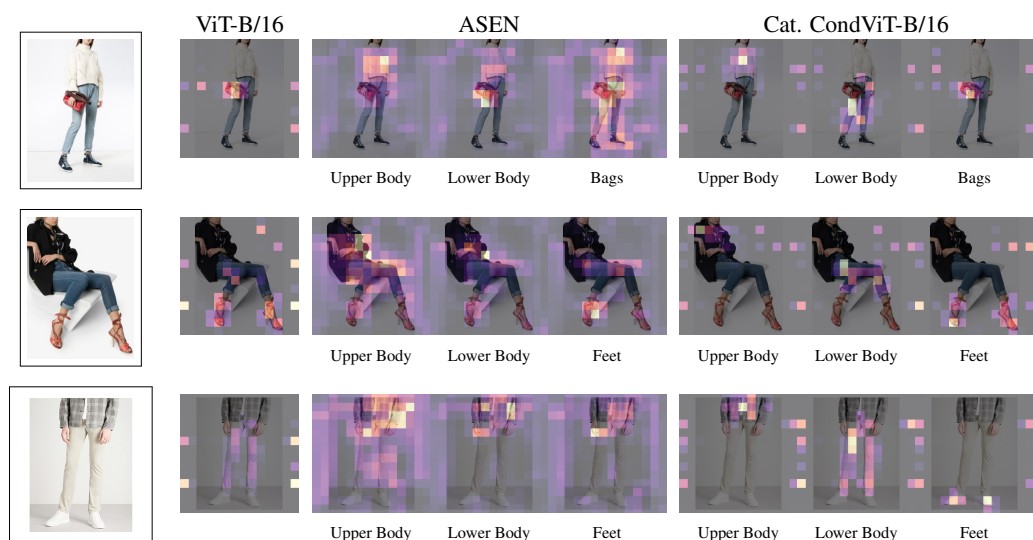

Figure 10: Attention maps. For ViT-B/16 and CondViT-B/16, we display the maximum attention from the CLS token to the image tokens across all heads in the last layer, and observe sparse maps. For ASEN, we display the attention returned by the Spatial Attention module of the global branch, and observe more diffuse maps. All maps are normalized to [0-1].

Figure 11 shows the patch features extracted by our models with principal component analysis (PCA) computed on all image tokens in the last layer of our CondViT-B/16 model across the test queries. Similarly to Oquab et al. [39], we find that applying a threshold on the first component enables effective separation of the background from the foreground. Intriguingly, we observe that employing a higher threshold not only accomplishes the aforementioned separation but also yields cleaner visualizations by isolating the conditionally selected object. We also observe instances where the network encounters difficulties in detecting the referenced object, resulting in a notable absence of tokens surpassing the established threshold.

## B.3 Textual Conditioning — Failure Cases

We finally present limitations of our textual CondViT-B/16 in Fig. 12. Firstly, when faced with failure in identifying the referenced object, our model resorts to selecting the salient object instead.

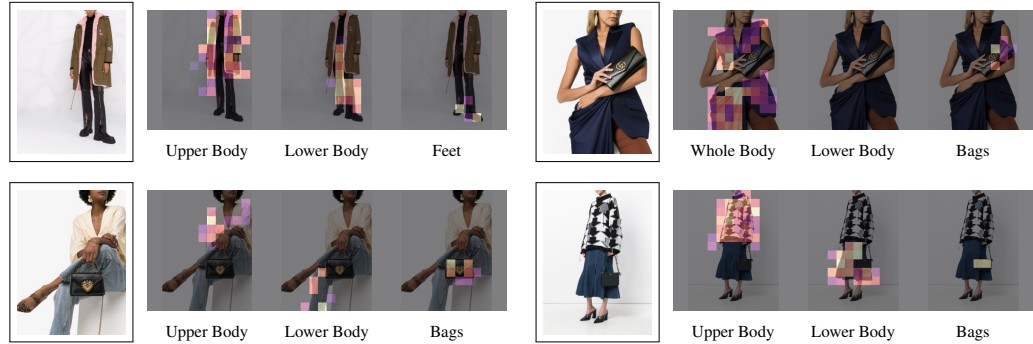

Figure 11: Visualization of the thresholded first component of image tokens in our CondViT-B/16. This component enables separation of the background, foreground, and focused object.

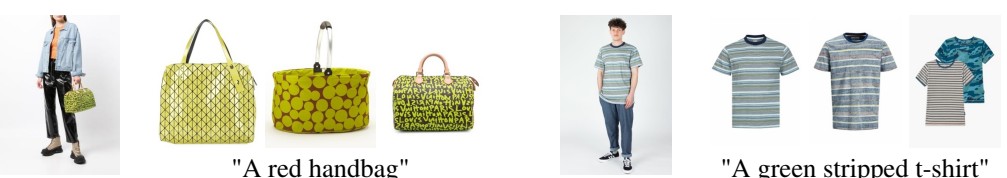

"A red handbag"          "A green stripped t-shirt"

(a) Top-3 retrieval for queries trying to modify color of an item. We find such modifications to be mostly ignored by the model.

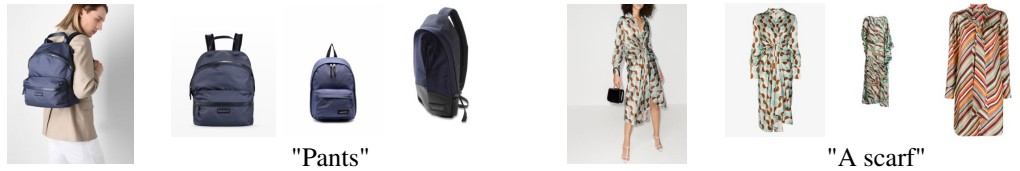

"Pants"          "A scarf"

(b) Top-3 retrieval for missed queries. For hard queries, or queries about an item not represented in the picture we find a tendency to default to the salient item in the image.

Figure 12: Retrieved items showing failure cases of our textual CondViT-B/16. (a) shows that the network disregards color clues. (b) shows that the network defaults to the salient item when the query is too hard or not represented.

Additionally, our model ignores queries with color or texture modifications, returning objects as depicted in the query image.

## B.4   Retrieval Examples

In this section, we show additional results for our categorical CondViT-B/16 and its textual variant trained with BLIP2 [29] captions. We use test query images and the full test gallery with 2M distractors for the retrieval. Each query in the test set is exclusively associated with a single item. However, it should be noted that the we do not necessarily query for this item, so the queried product might not be in the gallery. Nevertheless, owing to the presence of 2M distractors, most queries can retrieve multiple viable candidates.

Fig. 13 shows that our categorical CondViT is able to extract relevant features across a wide range of clothing items, and propose a coherent retrieval especially for the main categories. There is still room for improvement on images depicting rare training categories like *Waist*, *Hands*, *Head* or *Neck*, and rare poses.

Fig. 14 presents improvements brought by textual conditioning captions over categorical conditioning. Using text embeddings allows for more natural querying, thanks to the robustness of our model to irrelevant words. However, this robustness comes at the cost of ignoring appearance modifications.

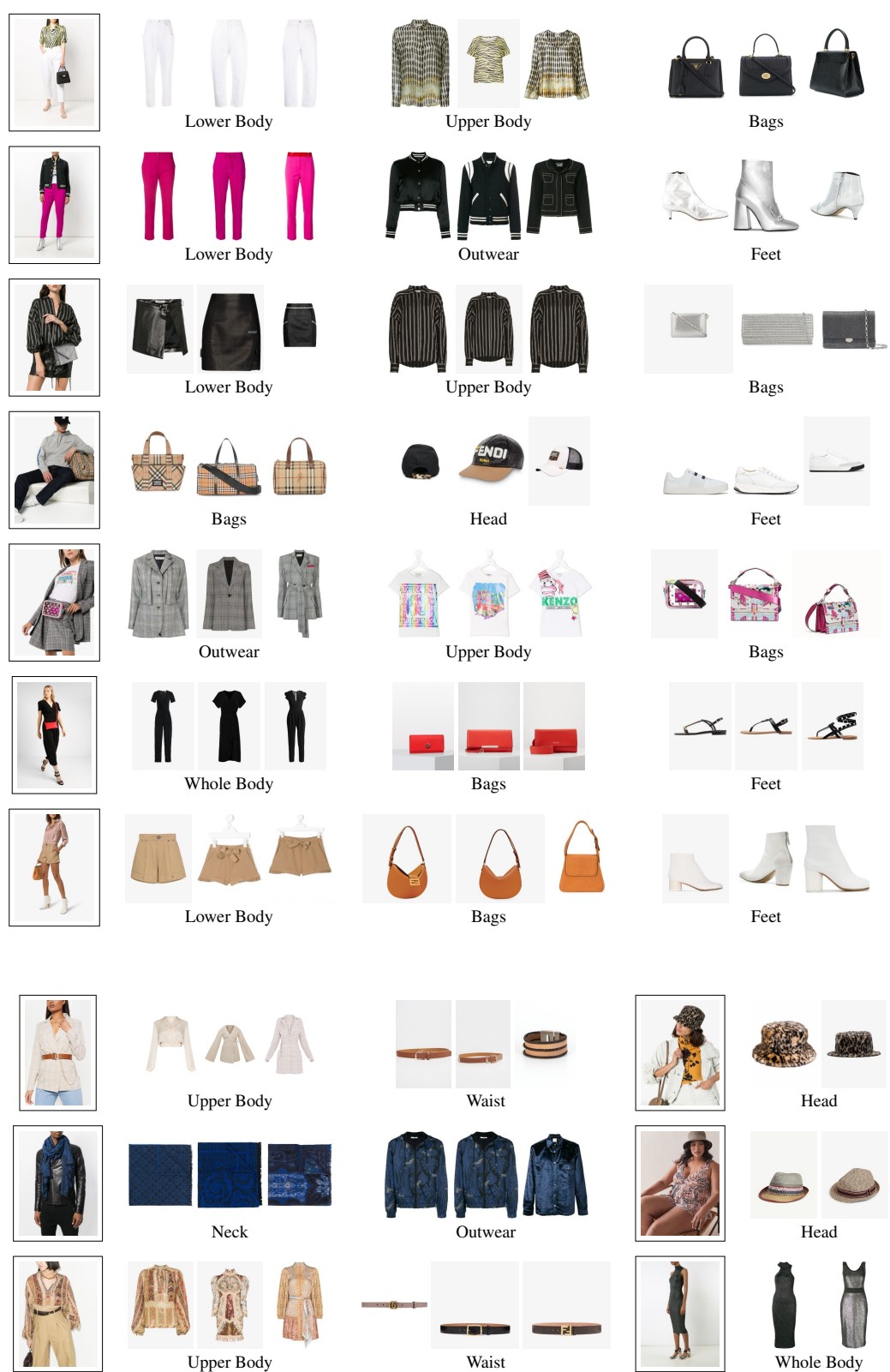

Figure 13: Qualitative results of our Conditional ViT-B/16 on LRVS-F test set.

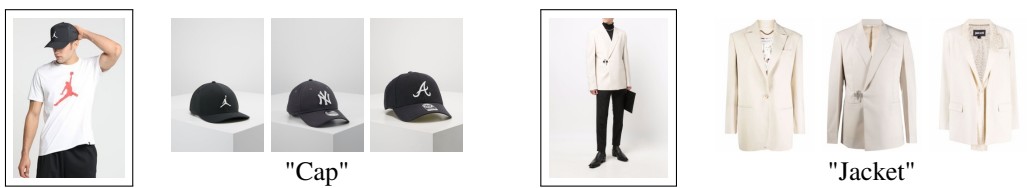

(a) Top-3 retrieval for normal user queries. Even though the BLIP2 captions were more detailed, using a single word as a query produces the expected result.

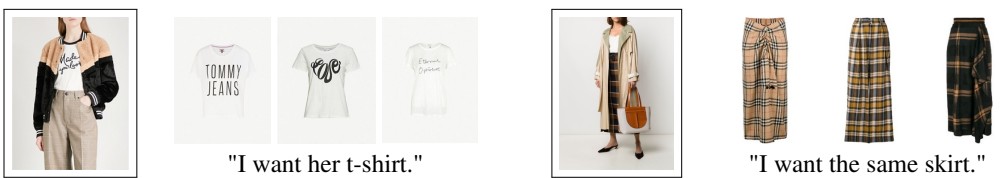

(b) Top-3 retrieval for noisy user queries. Our model is robust to expression of user intent and can focus on the designated object.

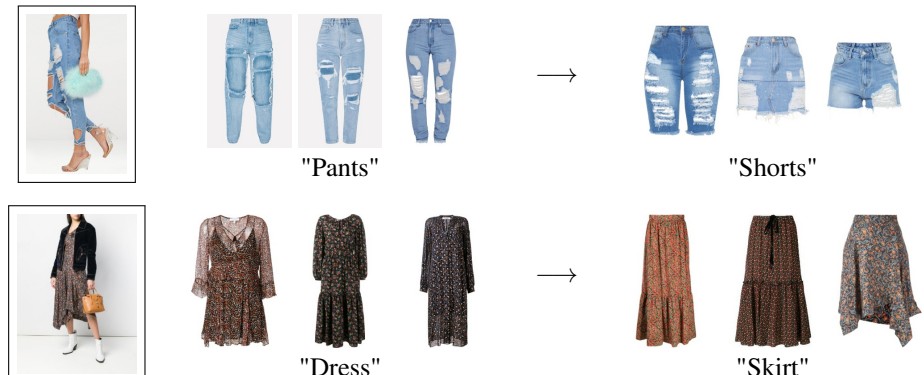

(c) Top-3 retrieval for queries with item modifications. In some circumstances, a textual query can influence the result to slightly modify the type of retrieved items, *e.g.* exchanging shorts and pants or skirts and dresses.

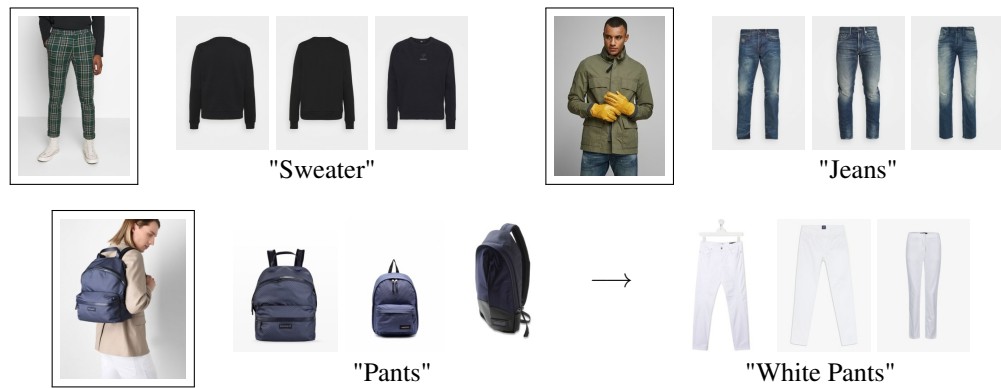

(d) Top-3 retrieval for out-of-frame items. If the network fails, we find that precising the query can help.

Figure 14: Retrieved items for queries in LRVS-F test set with our textual CondViT-B/16. (a) shows results for normal, concise use. (b) shows results with more verbose queries. (c) shows queries influencing the type of results. (d) show results for out-of-frame items.

# C Datasheet

## C.1 Motivations

Q1. **For what purpose was the dataset created?** *Was there a specific task in mind? Was there a specific gap that needed to be filled? Please provide a description.*

This dataset has been created to provide public training data and a benchmark for the Referred Visual Search (RVS) task, for research purposes. The task is new in academic research, and thereby no other dataset existed to tackle it.

Q2. **Who created the dataset (e.g., which team, research group) and on behalf of which entity (e.g., company, institution, organization)?**

The dataset was created by Simon Lepage, Jérémie Mary and David Picard, on behalf of the CRITEO AI Lab and ENPC.

Q3. **Who funded the creation of the dataset?** *If there is an associated grant, please provide the name of the grantor and the grant name and number.*

CRITEO AI Lab.

Q4. **Any other comments ?**

No.

## C.2 Composition

Q5. **What do the instances that comprise the dataset represent (e.g., documents, photos, people, countries)?** *Are there multiple types of instances (e.g., movies, users, and ratings; people and interactions between them; nodes and edges)? Please provide a description.*

Instances of this dataset are URLs from online catalogs of fashion retailers. The associated images depict fashion products isolated and in context. Some products are associated with a synthetic product identifier.

Q6. **How many instances are there in total (of each type, if appropriate)?**

In total, there are :

- 299,585 target simple images
- 486,995 complex images
- 59,938 partial complex images
- 2,099,555 additional simple images, not linked to any product, that serve as distractors.

Q7. **Does the dataset contain all possible instances or is it a sample (not necessarily random) of instances from a larger set?** *If the dataset is a sample, then what is the larger set? Is the sample representative of the larger set (e.g., geographic coverage)? If so, please describe how this representativeness was validated/verified. If it is not representative of the larger set, please describe why not (e.g., to cover a more diverse range of instances, because instances were withheld or unavailable).*

Our dataset is merely a small fashion subset of LAION-5B, which is itself a subset of the CommonCrawl dataset. We only selected a small amount of retailers and brands, mostly with European and American influence. As such, it is only a sample of fashion images, and is not representative of retailers and brands from other geographical areas.

Q8. **What data does each instance consist of? "Raw" data (e.g., unprocessed text or images) or features?** *In either case, please provide a description.*

Instances of the dataset are URLs of images, accompanied by various metadatas. Among them, their widths, heights, probabilities of containing a watermark, probabilities of being NSFW, associated texts (translated to english when needed) and original languages all originate from the LAION-5B dataset, and we refer the reader to this dataset for additional information. They are not used in the benchmark but we report them for ease of use and safety.

We added multiple synthetic labels to the images. First, a type, COMPLEX when the image depicts a scene, with a model, SIMPLE when it is an isolated product. There also exist a PARTIAL COMPLEX category, for scene images that are zoomed-in and do not contain the entire product. Second, a product identifier, allowing to group images depicting the same product. Each simple target image is further described by a category, following the taxonomy described in this paper, and 2 BLIP2-FlanT5XL captions.

Q9. **Is there a label or target associated with each instance?** *If so, please provide a description.*

We added categories and captions associated with each simple training image, but they are intended to be used as inputs to the models. The product identifier could be seen as a target as we propose a product retrieval task.

Q10. **Is any information missing from individual instances?** *If so, please provide a description, explaining why this information is missing (e.g., because it was unavailable). This does not include intentionally removed information, but might include, e.g., redacted text.*

Yes, complex images often depict multiple objects, but are linked with only one product in this dataset. They are registered in online fashion catalogs with the intent to showcase a specific product, and as such we were not able to extract more information.

Q11. **Are relationships between individual instances made explicit (e.g., users' movie ratings, social network links)?** *If so, please describe how these relationships are made explicit.*

Yes, we provide a synthetic product identifier for each image (exluding distractors), allowing to group simple and complex images depicting the same product.

Q12. **Are there recommended data splits (e.g., training, development/validation, testing)?** *If so, please provide a description of these splits, explaining the rationale behind them.*

Yes. We selected 400 products and 99,541 distractors to create a validation set. We also selected 2,000 products and 2,000,014 distractors to create a large test set. We selected the products so that their category distribution roughly match their distribution in the training set.

Q13. **Are there any errors, sources of noise, or redundancies in the dataset?** *If so, please provide a description.*

We created most of the new labels synthetically, using classifiers and captioners, so they contain some noise. However, by randomly sampling images and manually verifying their labels, we find an empiric error rate of 1/1000 for training complex images, 0/1000 for training simple images, and 3/1000 for distractors. Regarding the categories, we find an empiric error rate of less than 1%, with the confusions mostly stemming from semantically similar categories and images where object scale was ambiguous in isolated settings (long shirt against short dress, wristband against hairband).

The BLIP2 captions that we provide are of good quality and increase the mean CLIP similarity with the image of +7.4%. However, as synthetic captions, they are not perfect and sometimes contain hallucinations.

There are some redundancies in the distractors sets.

Q14. **Is the dataset self-contained, or does it link to or otherwise rely on external resources (e.g., websites, tweets, other datasets)?** *If it links to or relies on external resources, a) are there guarantees that they will exist, and remain constant, over time; b) are there official archival versions of the complete dataset (i.e., including the external resources as they existed at the time the dataset was created); c) are there any restrictions (e.g., licenses, fees) associated with any of the external resources that might apply to a future user? Please provide descriptions of all external resources and any restrictions associated with them, as well as links or other access points, as appropriate.*

No, the dataset relies on external links to the World Wide Web. We are unable to offer any guarantees of the existence of the images over time. We do not own the rights of these images, and as such do not provide any archival version of the complete dataset. These

copyrights might contains restriction about the images use. We encourage any user of the dataset to inquire about these copyrights.

Q15. **Does the dataset contain data that might be considered confidential (e.g., data that is protected by legal privilege or by doctor–patient confidentiality, data that includes the content of individuals' non-public communications)?** *If so, please provide a description.*

No. This dataset only contains samples from online fashion catalogs, and as such does not contain any confidential or personal data.

Q16. **Does the dataset contain data that, if viewed directly, might be offensive, insulting, threatening, or might otherwise cause anxiety?** *If so, please describe why.*

No. This dataset contains samples from online fashion catalogs, that result from professional photoshoots with the objective to be as appealing a possible to a large amount of customers.

Q17. **Does the dataset relate to people?** *If not, you may skip the remaining questions in this section.*

Models are present in the complex images. However, the sole focus of our dataset is the fashion items they are wearing, and most of the images are isolated objects. It does not contain any private or personal information.

Q18. **Does the dataset identify any subpopulation (e.g., by age, gender)?** *If so, please describe how these subpopulations are identified and provide a description of their respective distributions within the dataset.*

No, the dataset does not contain any metadata allowing to identify any subpopulation.

Q19. **Is it possible to identify individuals (i.e., one or more natural persons), either directly or indirectly (i.e., in combination with other data) from the dataset?** *If so, please describe how.*

It might be possible to identify models using facial recognition, but it would require external data.

Q20. **Does the dataset contain data that might be considered sensitive in any way (e.g., data that reveals racial or ethnic origins, sexual orientations, religious beliefs, political opinions or union memberships, or locations; financial or health data; biometric or genetic data; forms of government identification, such as social security numbers; criminal history)?** *If so, please provide a description.*

No.

Q21. **Any other comments ?**

No.

## C.3   Collection Process

Q22. **How was the data associated with each instance acquired?** *Was the data directly observable (e.g., raw text, movie ratings), reported by subjects (e.g., survey responses), or indirectly inferred/derived from other data (e.g., part-of-speech tags, model-based guesses for age or language)? If data was reported by subjects or indirectly inferred/derived from other data, was the data validated/verified? If so, please describe how.*

The initial data was acquired from LAION-5B, a subset of CommonCrawl. Please refer to their work for details about this initial data acquisition. The additional labels were synthetically generated by deep neural networks, based on manually annotated data, and a pretrained captioner.

Q23. **What mechanisms or procedures were used to collect the data (e.g., hardware apparatus or sensor, manual human curation, software program, software API)?** *How were these mechanisms or procedures validated?*

We manually curated domains and manually designed regular expressions to extract product identifiers from the URLs. The additional labels and captions are synthetic. We validated the quality of the labels by measuring accuracy on random samples, and the captions with a

CLIP similarity. Most of the process was done on a single CPU node, with the exception of the deep learning models which were run on two GPUs.

Q24. **If the dataset is a sample from a larger set, what was the sampling strategy (e.g., deterministic, probabilistic with specific sampling probabilities)?**

The dataset is a sample from LAION. The URLs were chosen based on a list of curated fashion retailers domains, selected for the quality of their images and their use of simple and complex images to showcase a product.

Q25. **Who was involved in the data collection process (e.g., students, crowdworkers, contractors) and how were they compensated (e.g., how much were crowdworkers paid)?**

The authors were the only persons involved in this data collection process.

Q26. **Over what timeframe was the data collected? Does this timeframe match the creation timeframe of the data associated with the instances (e.g., recent crawl of old news articles)?** *If not, please describe the timeframe in which the data associated with the instances was created.*

The data was collected from LAION and annotated at the beginning of 2023. This timeframe does not match the timeframe associated with the instances. The LAION-5B dataset has been created between September 2021 and January 2022, based on CommonCrawl. CommonCrawl itself is a collection of webpages started in 2008. However, it is impossible to know for certain how far the data stretches, as the websites might include older pictures.

Q27. **Were any ethical review processes conducted (e.g., by an institutional review board)?** *If so, please provide a description of these review processes, including the outcomes, as well as a link or other access point to any supporting documentation.*

The dataset is currently under review.

Q28. **Does the dataset relate to people?** *If not, you may skip the remaining questions in this section.*

The dataset contains some images of fashion models, but it does not contain any personal data and focuses on objects.

Q29. **Did you collect the data from the individuals in question directly, or obtain it via third parties or other sources (e.g., websites)?**

No, we obtained it from LAION-5B.

Q30. **Were the individuals in question notified about the data collection?** *If so, please describe (or show with screenshots or other information) how notice was provided, and provide a link or other access point to, or otherwise reproduce, the exact language of the notification itself.*

Please refer to LAION-5B.

Q31. **Did the individuals in question consent to the collection and use of their data?** *If so, please describe (or show with screenshots or other information) how consent was requested and provided, and provide a link or other access point to, or otherwise reproduce, the exact language to which the individuals consented.*

Please refer to LAION-5B.

Q32. **If consent was obtained, were the consenting individuals provided with a mechanism to revoke their consent in the future or for certain uses?** *If so, please provide a description, as well as a link or other access point to the mechanism (if appropriate).*

Please refer to LAION-5B.

Q33. **Has an analysis of the potential impact of the dataset and its use on data subjects (e.g., a data protection impact analysis) been conducted?** *If so, please provide a description of this analysis, including the outcomes, as well as a link or other access point to any supporting documentation.*

This dataset and LAION 5B have been filtered using CLIP-based models. They inherit various biases contained in their original training set. Furthermore, the selected domains

in this work only represent European and American fashion brands, and do not provide a comprehensive view of worldwide fashion.

Q34. **Any other comments ?**

No.

### C.4 Preprocessing / Cleaning / Labeling

Q35. **Was any preprocessing/cleaning/labeling of the data done (e.g., discretization or bucketing, tokenization, part-of-speech tagging, SIFT feature extraction, removal of instances, processing of missing values)?** *If so, please provide a description. If not, you may skip the remainder of the questions in this section.*

We started with a list of fashion domains with images of good quality, and extracted the corresponding images from LAION. We then trained a first classifier with an active learning procedure to classify the complexity of the obtained images. A second classifier was trained in the same way to classify the categories of the simple images, and captions were added using BLIP2-FlanT5XL.

We extracted product identifiers from the URLs, and kept products that were represented at least in a simple and a complex images. The discarded images, and those for which we couldn't extract any identifiers, are used as distractors.

We used LSH and KNN indices to remove duplicates among products, and between the products and the distractors in the validation and test sets.

Please refer to Section. 3.1 and Appendix A.3 for additional details.

Q36. **Was the "raw" data saved in addition to the preprocessed/cleaned/labeled data (e.g., to support unanticipated future uses)?** *If so, please provide a link or other access point to the "raw" data.*

The "raw" data is LAION-5B.

Q37. **Is the software used to preprocess/clean/label the instances available?** *If so, please provide a link or other access point.*

No, apart from img2dataset that we used to download the images. Many critical parts in the process were manually supervised, such as extracting product identifiers for each domain, labeling during the active learning process, and checking the duplicates returned by the similarity search.

Q38. **Any other comments ?**

No.

### C.5 Uses

Q39. **Has the dataset been used for any tasks already?** *If so, please provide a description.*

This is the first time that the LRVS-F dataset is used. We use it to study the Referred Visual Search task. The goal of this task is to retrieve a specific object among a large database of distractors given a complex image and additional referring information (category or text).

Q40. **Is there a repository that links to any or all papers or systems that use the dataset?** *If so, please provide a link or other access point.*

No.

Q41. **What (other) tasks could the dataset be used for?**

The dataset could be used for other fashion-related tasks, like fashion generation or virtual try-on.

Q42. **Is there anything about the composition of the dataset or the way it was collected and preprocessed/cleaned/labeled that might impact future uses?** *For example, is there anything that a future user might need to know to avoid uses that could result in unfair treatment of individuals or groups (e.g., stereotyping, quality of service issues) or other*

*undesirable harms (e.g., financial harms, legal risks) If so, please provide a description. Is there anything a future user could do to mitigate these undesirable harms?*

Our dataset only contains large European and American fashion retailers. As such, it does not reflect the diversity of fashion cultures across the globe, and future users should not expect it to generalize to other geographical areas or specific localities.

Q43. **Are there tasks for which the dataset should not be used?** *If so, please provide a description.*

This dataset is for research purpose only, and contains biases. We warn any user against using it as-is outside of this context, and emphasize that results obtained on this dataset cannot be expected to generalize to any culture without proper bias study.

Q44. **Any other comments?**

As the images still belong to their respective owner, we only release this dataset for research purpose. We encourage anyone willing to use the images for commercial use to verify their copyright state with their respective rightholders.

Furthermore, we encourage users to respect opt-out policies, through the use of dedicated tools like img2dataset and SpawningAI.

## C.6    Distribution

Q45. **Will the dataset be distributed to third parties outside of the entity (e.g., company, institution, organization) on behalf of which the dataset was created?** *If so, please provide a description.*

Yes, the dataset is open-source and freely accessible.

Q46. **How will the dataset be distributed (e.g., tarball on website, API, GitHub)?** *Does the dataset have a digital object identifier (DOI)?*

The dataset will be available as a collection of parquet files containing the necessary metadata. It will have a DOI.

Q47. **When will the dataset be distributed?**

It is already available.

Q48. **Will the dataset be distributed under a copyright or other intellectual property (IP) license, and/or under applicable terms of use (ToU)?** *If so, please describe this license and/or ToU, and provide a link or other access point to, or otherwise reproduce, any relevant licensing terms or ToU, as well as any fees associated with these restrictions.*

We release our data under the CC-BY-NC-4.0 license.

Q49. **Have any third parties imposed IP-based or other restrictions on the data associated with the instances?** *If so, please describe these restrictions, and provide a link or other access point to, or otherwise reproduce, any relevant licensing terms, as well as any fees associated with these restrictions.*

We only own the synthetic metadata that we release. The attributes of the dataset that originate from LAION-5B belong to LAION and are distributed under a CC-BY 4.0 license. We do not own the copyright of the images and original alt texts.

Q50. **Do any export controls or other regulatory restrictions apply to the dataset or to individual instances?** *If so, please describe these restrictions, and provide a link or other access point to, or otherwise reproduce, any supporting documentation.*

No.

Q51. **Any other comments ?**

No.

## C.7    Maintenance

Q52. **Who will be supporting/hosting/maintaining the dataset?**

The dataset is hosted at https://huggingface.co/datasets/Slep/LAION-RVS-Fashion.

Q53. **How can the owner/curator/manager of the dataset be contacted (e.g., email address)?**

The owner of the dataset can be contacted through the dataset's HuggingFace space.

Q54. **Is there an erratum?** *If so, please provide a link or other access point*

There is no erratum as this is the initial release. If need be, we will update the dataset repository.

Q55. **Will the dataset be updated (e.g., to correct labeling errors, add new instances, delete instances)?** *If so, please describe how often, by whom, and how updates will be communicated to users (e.g., mailing list, GitHub)?*

We do not plan to update the dataset, as it contains a benchmark and we want the results to stay comparable across time.

Q56. **If the dataset relates to people, are there applicable limits on the retention of the data associated with the instances (e.g., were individuals in question told that their data would be retained for a fixed period of time and then deleted)?** *If so, please describe these limits and explain how they will be enforced.*

The dataset does not relate to people. It does not contain personal or private information.

Q57. **Will older versions of the dataset continue to be supported/hosted/maintained?** *If so, please describe how. If not, please describe how its obsolescence will be communicated to users.*

There is currently no older version of this dataset. If changes must be made, the updates will be applied on the hosting page but history of changes will stay available.

Q58. **If others want to extend/augment/build on/contribute to the dataset, is there a mechanism for them to do so?** *If so, please provide a description. Will these contributions be validated/verified? If so, please describe how. If not, why not? Is there a process for communicating/distributing these contributions to other users? If so, please provide a description.*

We do not plan on supporting extensions to this dataset as it is intended to be a benchmark and results must stay comparable across time. However we do encourage the creation of similar datasets across new verticals, to extend the field of Referred Visual Search.

Q59. **Any other comments ?**

No.