# OpenReview forum: "LRVS-Fashion: Extending Visual Search with Referring Instructions"
_NeurIPS.cc/2024/Datasets_and_Benchmarks_Track — Submitted to NeurIPS 2024 Track Datasets and Benchmarks_

### Official Review · Reviewer_rUM1 · 2024-06-30
**Review of "LRVS-Fashion: Extending Visual Search with Referring Instructions"**

**Rating:** 6
**Confidence:** 4
**Clarity:** The paper is fairly well written, alt…

**Review:**

This paper presents Referred Visual Search (RVS) in the context of fashion, introduces a new dataset, and proposes a method to extract referred embeddings for this task.
Strengths:
1.	The paper introduces a new dataset explicitly designed for RVS, providing a platform for future research and benchmarking.
2.	The authors provide a detailed description of the dataset's information and construction process.
3.	The authors offer comprehensive code for readers to replicate the experiments.
Weaknesses:
1.	The specific meaning of the latent information \( c_t \) is not clearly explained in the method description. The asymmetrical setup is shown in the method diagram but not adequately reflected in the text.
2.	Figure 1 demonstrates that a single complex image can effectively retrieve four different simple objects under four different reference conditions. However, in the test set, it seems that each complex image only uses one reference instruction for the final quantitative results calculation. This may not adequately reflect the model's robustness.

**Strengths:**

The development of the LRVS-Fashion dataset, comprising 272k fashion products with 842k images, is a substantial contribution. This dataset, explicitly designed for RVS, provides a platform for future research and benchmarking. By focusing on a practical problem with real-world implications, the paper is likely to attract significant attention from both researchers and practitioners.

**Additional Feedback:**

It seems that all the images are free from background interference. If the images were in more complex scenes, such as in the Exact Street2Shop Dataset[1], would the proposed weakly-supervised method still be effective?

[1]M. H. Kiapour, X. Han, S. Lazebnik, A. C. Berg and T. L. Berg, "Where to Buy It: Matching Street Clothing Photos in Online Shops," 2015 IEEE International Conference on Computer Vision (ICCV), Santiago, Chile, 2015, pp. 3343-3351, doi: 10.1109/ICCV.2015.382.

**Correctness:**

The authors employ an asymmetrical setup, which is reasonable. By adding a large number of distractors to the test set, the evaluation becomes more realistic.

**Documentation:**

The paper provides sufficient details on the methodology and experimental setup, ensuring that the readers can fully understand the approach. Additionally, the authors have released the code, enabling others to replicate the experiments and validate the results.

**Ethics:**

There are no ethical concerns.

**Limitations:**

The system may ignore the conditioning if the desired item is small or absent in the database. This means that the retrieval performance can be compromised if the target object is not prominently visible or if it does not exist in the database.

**Opportunities For Improvement:**

The paper could further enhance the retrieval performance for small objects. This could potentially involve refining the conditioning mechanism or incorporating advanced techniques for detecting and emphasizing minor objects within complex scenes.

**Relation To Prior Work:**

The paper introduces a Referring task for fashion content retrieval, which is similar to open-vocabulary object detection. It offers a comprehensive overview and references relevant work in the field of image retrieval. This task requires processing both image and text information, and the authors provide a concise introduction to vision-language models. Additionally, they explain the differences between this new task and previous tasks, highlighting the unique challenges and approaches involved in integrating visual and textual data for more precise retrieval.

**Summary And Contributions:**

This paper presents Referred Visual Search (RVS), a task for retrieving fashion images based on user instructions. The authors constructed a dataset composed of pairs of complex and isolated images, which is a subset of LAION 5B, and enriched it with metadata using BLIP2. They also demonstrated an effective weakly-supervised training method that achieved superior accuracy compared to a strong detection-based baseline and existing published works. Extensive experiments were conducted, and a detailed description of the dataset is provided.

---

> ### Author Rebuttal · Authors · 2024-08-14
>
> We thank the reviewer for their positive feedback and for recognizing the utility of our dataset. We address their main concerns below.
>
> > **The specific meaning of the latent information ( c_t ) is not clearly explained**
>
> In our case, the target image is an isolated product, and we only defined category-based similarities, so we do not need to condition the target side (see the ablation study in Section B.1, Table 5). However, it is possible to define RVS settings in which both sides are conditioned (e.g., using conditioned complex images as both query and target). Therefore, we chose to keep a generic definition in the formalization of the task.
>
> > **It seems that each complex image only uses one reference instruction**
>
> Yes, this is due to the nature of the data, as each image was linked to a single product identifier. We report the performance of unconditional ViT models in Table 2 to provide a reference metric regarding this bias, as the model could potentially infer the object of interest from the framing of the picture.
>
> > **The paper could further enhance the retrieval performance for small objects.**
> > **The system may ignore the conditioning if the desired item is small or absent in the database.**
>
> Yes, these are common shortcomings of visual search systems, and ours is no different. We encourage future work on RVS to focus on these issues, as our goal with this paper was primarily to provide a clean and useful dataset and to propose the first baselines.
>
> > **If the images were in more complex scenes, such as in the Exact Street2Shop Dataset, would the proposed weakly-supervised method still be effective?**
>
> Our hypothesis is that the method would remain effective in more complex scenes, as the task can be understood as learning to conditionally ignore or focus on specific parts of the image. If trained on complex images, the network would likely learn to disregard elements that do not contribute to lowering the loss.
> We have preliminary experiments and results on private datasets that align with those presented in the paper, suggesting that the method is robust to this type of perturbation.
>
> However, please note that gathering and releasing more realistic in-the-wild pictures taken by users can be subject to ethical concerns, regarding privacy for example.

---

### Official Review · Reviewer_UHox · 2024-07-24
**Reviewer UHox**

**Rating:** 5
**Confidence:** 3
**Correctness:** This is fine.
**Clarity:** This is clear enough.

**Review:**

The paper would be more suited to the main track rather than the D&B track. Although it proposes a new derivative dataset, little analysis is provided, and benchmarks are not comprehensive enough. The main focus of the paper seems to be on the proposed task and how to design ViT for it.

**Strengths:**

* The task of retrieving a specific fashion product from a query with multiple products in the image is an interesting, practical and useful task.
* Figures in the paper explain well the problem and the approach.

**Additional Feedback:**

It seems like the authors used the wrong template. The line spacing seems different from other submissions to this year D&B track, allowing more text to be included in the submission.

**Documentation:**

* No code is provided to reproduce the derivative dataset collection
* Code is provided to reproduce the proposed model
* Dataset is hosted on huggingface with a license
* An informative datasheet is also provided

**Ethics:**

I am hesitant to flag for an ethics review. I will flag it, to be on the safe side.

The paper proposes a derivative from LAION-5B, which is a dataset that was recently retracted for containing child abuse images. In that regard, there should not be any derivative being made from LAION-5B given the high risks associated with it. There is no comment from the authors around this issue, so there exist a non-zero possibility for the proposed dataset to contain such images.

Furthermore, it seems that such crawled images contain faces of people who have not given their consent to be part of the dataset.

**Limitations:**

This is not sufficient. The paper should discuss the lack of complex background in images (see above for details), as well as the fact that the dataset derives from LAION-5B, which is a highly problematic dataset (see below for details).

**Opportunities For Improvement:**

* Why rename an existing task?
  * The paper claims to introduce a new task, and calls it "Referred Visual Search". Though throughout the introduction and the related works, the paper consistently mentions that the task already exists in the literature. Why the need to rename an existing task, as this would add more confusion.

* Mismatch between proposed use case and the proposed dataset
  * The proposed dataset seems to contain images with a uniform background. How would this translate to in-the-wild images which would be the proposed use case by "Google, Amazon, and Pinterest"? How big would be the domain gap? Would the proposed approach without bounding box proposals still relevant?

* No quality check on the generated meta-data
  * Fashion categories and captions are generated with an LLM. The paper even mentions having hallucination. There is no process to actually check the quality of such generated outputs while they are used for the retrieval task.

* Dataset construction is under-documented
  * It is unclear how to reproduce the proposed dataset. Indeed, it is a derivative from LAION-5B, but we do not know how fashion products were derived from it.

* Benchmarks and their analyses are insufficient
  * The related work lists plenty of alternative methods, yet only 3 existing models are evaluated. Why so few models are compared with the proposed methods?
  * When evaluating CLIP4CIR [2], it appears that the performance is in the same ballpark than the proposed model. What advantage is there in the proposed model, besides the marginal improvement?
  * The proposed dataset seems quite easy, as when there is little distraction, the recall@1 is already very high. What are then challenges associated with it?
  * While there exist *category* and *caption* annotations, there is no proposal on using both to improve retrieval. They are only used separately.

**Relation To Prior Work:**

The introduction would benefit from having more references to support claims being made. For example, there is to reference to the following claims:
* L18-19 "the fact that wo images can be considered in various ways, leading to ambiguity"
* L26 "numerous industry leaders have adopted this tactic"

Otherwise, the related work section is sufficient.

**Summary And Contributions:**

The paper addresses fashion product image retrieval. The idea is to have a query image of a full-body image with multiple garments, and retrieve image of a specific garment. To achieve this, the paper introduces a derivative dataset from LAION-5B and proposes a ViT model to integrate the category or caption for retrieving the specific fashion product images.

---

> ### Author Rebuttal · Authors · 2024-08-14
>
> We thank the reviewer for their feedback and address their main concerns below.
>
> > **The paper would be more suited to the main track rather than the D&B track**
>
> The goal of the paper is to introduce our dataset to encourage research on RVS. Most of the paper is dedicated to the construction of the dataset (Section 3.1 and Appendix A.3), its composition and quality (Section 3.2 and Appendix A.1/A.4), discussions on ethics and biases (Appendix A.5 and C), defining evaluation metrics (Section 3.3), and evaluating baselines for the task (Section 5.2). The remaining sections formalize the task and propose a tailored ViT as a modern and easily reproducible baseline for future work.
>
> We release the dataset, evaluation code, and a public leaderboard to make it a valuable benchmark for the community (along with training code and models).
>
> > **The paper claims to introduce a new task, and calls it "Referred Visual Search". Though throughout the introduction and the related works, the paper consistently mentions that the task already exists in the literature. Why the need to rename an existing task, as this would add more confusion.**
>
> We do not claim to introduce a new task; if a sentence gives this impression, please point it out so we can correct it. Similar tasks have indeed appeared in practical applications and in the literature over the past few years. However, we did not find any consensus regarding the name of the task, with most papers using their own implicit and imprecise terminology, such as variations of “similarity search” or “fashion embedding.” We believe this lack of consistency is detrimental to the field, as it makes searching for relevant papers more difficult.
>
> We propose the name “Referred Visual Search” to highlight its connection to well-known practical applications (Visual Search) and to other academic referring tasks like Referring Expression Segmentation or Comprehension.
>
> > **The proposed dataset seems to contain images with a uniform background. How would this translate to in-the-wild images which would be the proposed use case by "Google, Amazon, and Pinterest"? How big would be the domain gap? Would the proposed approach without bounding box proposals still relevant?**
>
> Our dataset focuses on professional pictures, so while some retailers do use “complex” backgrounds, the vast majority are indeed uniform. To create a dataset directly applicable to industrial use with optimal performances, we would need in-the-wild images paired with the products. However, acquiring and releasing such data publicly is challenging due to associated ethical concerns, such as privacy.
>
> We believe that our partial curation of LAION, focusing on professional pictures, is a good compromise that lowers the entry barrier for academic study of RVS, encouraging further development of the field while being clean and avoiding ethical issues.
>
> Our results on private datasets align with those presented in the paper, suggesting that the method is robust. We also received positive feedback when using our publicly available Gradio demo with in-the-wild pictures.
>
> > **Fashion categories and captions are generated with an LLM. The paper even mentions having hallucination. There is no process to actually check the quality of such generated outputs while they are used for the retrieval task.**
>
> Section 3.2 details the quality measures for each of the generated labels. For the categories, generated with a classifier trained using active learning, we observe very low error rates (<1%). For the captions, we measure an increase in CLIP similarity compared to the original alt texts. Empirically, we show samples in Figures 3 and 7, and during our experiments, we found the captions to be of overall good quality, as describing a simple item on a uniform background is straightforward for captioning models.
>
> > **It is unclear how to reproduce the proposed dataset. Indeed, it is a derivative from LAION-5B, but we do not know how fashion products were derived from it.**
>
> Section 3.1 and Appendix A.3 thoroughly detail the construction process. The derivation of fashion product identities from the URL is explained in lines 129 to 131. In short, we manually identified sections of the URLs corresponding to product identifiers for each fashion retailer and extracted them using regex.
>
> > **Why so few models are compared with the proposed methods?**
>
> Most of the models cited in the related work section lack open-source implementations and/or rely on localization labels, which are not available in our dataset.
>
> > **When evaluating CLIP4CIR [2], it appears that the performance is in the same ballpark than the proposed model. What advantage is there in the proposed model, besides the marginal improvement?**
>
> First, achieving good RVS performance with CLIP4CIR requires retraining it on our dataset, demonstrating the value of our contribution. Furthermore, our method is easily adaptable to categorical conditioning, which has not been demonstrated with CLIP4CIR. Finally, our training procedure is much simpler, with fewer hyperparameters, as CLIP4CIR requires a two-stage fine-tuning process and a new “Combiner” architecture. In the long run, simpler and more robust techniques tend to prevail.
>
> > **The proposed dataset seems quite easy, as when there is little distraction, the recall@1 is already very high. What are then challenges associated with it?**
>
> The interesting metrics for this dataset are those in large-scale retrieval settings (1M/2M), for two reasons:
> It requires more precise embeddings to distinguish between lookalike garments.
> E-commerce catalogs of large retailers typically contain millions of products.
> Therefore, developing model robustness to increasing amounts of distractors is a relevant challenge for this task.
>
> *— To be continued in an Official Comment below as we reach the character limit*

---

> ### Author Response · Authors · 2024-08-14
>
> *— Continued from the rebuttal*
>
> > **Introduction references : “The fact that two images can be considered similar in various ways”**
>
> We believe this is best explained with examples, as it is mostly common sense. Imagine a collection of photos depicting models wearing clothes in various poses. We could define different similarities between them based on whether a) the model is the same, b) the garments are the same, or c) the pose of the model and picture framing are the same. We could multiply the number of examples, but we are unsure of the added value.
>
> > **Introduction references : “Numerous industry leaders have adopted this tactic”**
>
> We cite three industry leaders using this tactic, which are our references for this claim. Google Lens, Amazon Lens, and Pinterest Lens all allow users to specify the object of interest in ambiguous image queries.
>
> > **There is no comment from the authors around this issue, so there exist a non-zero possibility for the proposed dataset to contain such images.**
>
> Appendix A.5 contains a brief discussion of this issue. To summarize, our dataset only contains images that appear in online fashion catalogs of well-known brands, and as such, it does not contain harmful or disturbing images. Please refer to the Ethical Review we received and our response for further discussion.
>
> > **It seems that the authors used the wrong template**
>
> We mistakenly used the template for NeurIPS D&B 2023 as a placeholder before the 2024 version was released and forgot to replace it. However, the only differences between them are the dates; the **spacing is the same**, and fixing this mistake **does not result in overflowing content.**

---

> > ### Comment · Reviewer_UHox · 2024-08-28
> > **Response**
> >
> > Thank you for the rebuttal. It partially addresses some of my concerns. I will upgrade my score but still think the paper is below the acceptance bar.
> >
> > In particular, a few points: the lack of comparisons with existing methods in visual search, or the insufficient arguments that "results on private datasets align with those presented in the paper" or "positive feedback when using our publicly available Gradio demo with in-the-wild pictures" which are not quantified or reproducible.

---

### Official Review · Reviewer_3aj4 · 2024-07-24
**introduces task of referred visual search, an accompanying dataset and method that outperforms strong baselines**

**Rating:** 7
**Confidence:** 4
**Clarity:** yes

**Review:**

Quality: This paper introduces a new comprehensive dataset, a rigous method, and detailed implementation description.

Clarity: The paper is well organized and effectively uses visual aids. The descriptions are largely easy to follow but there are some sections that were technically dense/required a couple passes to comprehend.

Originality: The paper presents a novel task definition and accompanying dataset and presents a method that outperforms strong baselines.

Significance: This papers dataset and method have meaningful practical applications in fashion search that I suspect practitioners can benefit from now. It is unclear if a similar scheme would transfer over in other domains.


Pros

- Comprehensive dataset
- Novel task with clear industry/practical applications
- Well structured / presented paper
- New method with reasonable rigor in comparing to strong baselines

Cons

- data collection method appears to completely rely on synthetic labels, would be interesting to hear commentary on potential issues here or some qualitative comparisons to human quality
- limited commentary on failure cases / areas where the new method may struggle

**Strengths:**

strengths

- introduction of new referred visual search task and an accompanying comprehensive / large dataset with distractors
- benchmark/dataset has potential to spur further advances in fashion search
- detailed descriptions and rigor in evaluating method against strong baselines

**Additional Feedback:**

- i wonder if having all the distractors come from simple images might make retrieval artificially easier, it would be interesting to hear more commentary on the distractor selection

**Correctness:**

Are the claims made in the submission correct? yes

If the submission is a dataset, it is constructed in a sound way? yes

If it is a benchmark, are the evaluation methods and experiment design appropriate and performed correctly? yes

**Documentation:**

For datasets, is there sufficient detail on data collection and organization, availability and maintenance, and ethical and responsible use? yes

For benchmarks, is there sufficient detail to support reproducibility? yes

**Ethics:**

not really.

i think main thing i would note on this front is that these images originate from different commercial websites, its unclear if this application fails within each companys terms of use, though i suspect this is fair use but i'm not a lawyer lol

**Limitations:**

ve the authors adequately addressed the limitations and potential negative societal impact of their work? yes

**Opportunities For Improvement:**

some limitations / improvements

- dataset collected with reliance on synthetic/generated labels, would be helpful to have some analysis / commentary on the quality of these generations
- for complex images it is unclear how many sub categories are typical per image, would be nice to have some commentary on distributions like this
- unclear if this methodology would transfer to other domains, or even different settings. for example given that fashion images online largely have neutral or simple backgrounds (even in the cases that are considered 'complex' in dataset), it would be interesting to have query images that are more realistic to what a person might take in daily life , perhaps images from like amazon reviews of product, and seeing how that both impacts synthetic categorization/caption generation as well as performance on downstream task by say generating potential queries from these user provided text reviews
- data appears to focus on european/american fashion, would be interesting to consider other locales / considerations for more diverse fashions. would be helpful to disclose distribution of domains where these images originate

**Relation To Prior Work:**

yes

**Summary And Contributions:**

This paper introduces a new image similarity search task of referred visual search in the domain of fashion, which aims to address problems inherit to fashion similarity search, an does so using a methodology that avoids need of using segmentation/object detection.

some key contributions:

- definition of new task that takes into consideration specific details a user may be interested in given a fashion image and category and/or textual caption describing aspect of interest
- the LAION-RVS-Fashion dataset and description of collection methodology
- a method that leverages these category/textual references that produces representations that do not need object localizations or segmentation masks
- an evaluation that shows utility of method on this task over strong baselines using clip / grounding dino / segment anything

---

> ### Author Rebuttal · Authors · 2024-08-14
>
> We thank the reviewer for their encouraging feedback, and address their main concerns below.
>
> > **dataset collected with reliance on synthetic/generated labels, would be helpful to have some analysis / commentary on the quality of these generations**
>
> Section 3.2 of the paper provides empirical measures of the quality of the generated labels. The accuracy of the categories is very high, with an error rate of less than 1%. The captions are simple and show improvement over the original alt-text in terms of CLIP similarity (see Fig. 7 in the appendix for examples).
>
> > **limited commentary on failure cases / areas where the new method may struggle**
>
> Appendix B.3 provides examples of failure cases for the textual model. In general, we observe expected failure modes for a visual search system: most occur when the target object is very small, heavily occluded, out of frame, or extremely similar to other items in the database (such as pairs of jeans).
>
> > **how many sub categories are typical per image**
>
> As each complex image is linked to a single product, we do not have ground truth to provide a precise answer. Empirically, most images contain at least three categories (Feet, Upper Body, Lower Body), and up to six in the most complex cases (adding Outwear, Bags, and Head, for example).
>
> > **given that fashion images online largely have neutral or simple backgrounds, it would be interesting to have query images that are more realistic to what a person might take in daily life , perhaps images from like amazon reviews of product, and seeing how that both impacts synthetic categorization/caption generation as well as performance on downstream task by say generating potential queries from these user provided text reviews**
>
> We agree that studying RVS with more realistic “daily life” data would be a very interesting area of improvement. However, scraping reviews (and using in-the-wild data in general) raises ethical issues related to privacy. We believe that our partial curation of LAION, focusing on professional data, provides an initial academic tool to study this task without taking such risks.
>
> Regarding the impact on our construction process, the synthetic categories and captions were generated using simple images of the product. Using more realistic, “complex” images as queries does not interfere with this process as long as each product is still associated with a simple image, which is very common in e-commerce.
>
> > **data appears to focus on european/american fashion, would be interesting to consider other locales / considerations for more diverse fashions. would be helpful to disclose distribution of domains where these images originate**
>
> Indeed, Appendix A.5 provides a brief overview of this bias. We hope that the RVS task will draw attention and motivate other teams to develop datasets that focus on different locales or verticals, such as furniture.
>
> > **Having all the distractors come from simple images might make the retrieval artificially easier**
>
> We believe the opposite is true. By including distractors from the same distribution as the target images, we make the relevant retrieval space denser, requiring more precise embeddings. Finding the correct garment among tens of similar garments (in terms of color, shape, and patterns) is far more challenging than among unrelated images.

---

### Author Rebuttal · Authors · 2024-08-14

We appreciate the thoughtful feedback from the reviewers on our paper introducing the LRVS-Fashion dataset for the Referred Visual Search task (RVS) and our CondViT approach.

We are pleased that the reviewers recognized the **clear practical applications** of this task (3aj4, UHox, rUM1) and acknowledged our dataset as a significant contribution (3aj4, rUM1), providing a **valuable benchmark** for future research. The rigor of our experiments (3aj4) and the **reproducibility of our results**, supported by the provided code (rUM1), were also well received. Additionally, the **clarity of our presentation** and visual aids (3aj4, UHox, rUM1) were appreciated.

Although concerns were raised about the dataset's focus on professional images, we believe this approach effectively balances realism with ethical considerations, facilitating research while addressing the challenges and ethical issues associated with "in-the-wild" data (zmMk).

We have addressed various points regarding the dataset's construction, quality, and biases in our responses and will incorporate these clarifications into the updated paper.

Thank you for your consideration.

---

### Decision · Program_Chairs · 2024-09-26

**Decision:**

Reject

**Comment:**

This submission received three ratings (7, 5 and 6), averaging 6, which is a very borderline score (the targeted rate 25% requires avg. 6.33), and one reviewer also pointed out the wrong template of 2023 instead of 2024, which could induce more space compared to the ordinary template of 2024. The reviewers' concerns mainly focus on label synthesis (without quality check), the renaming, and the insufficient analysis. After rebuttal, no reviewers champion this submission including the positive reviewer with rating 7. After carefully checking the concerns of all reviewers and the authors' rebuttal as well as the given rating bar towards acceptance, I think the concerns of the reviewer  UHox greatly challenges the submission and requires the substantial clarification or modification, and thus suggest the rejection. Hope the comments of all reviewers can be helpful to improve the submission.